# Structural Failure of the Cohesive Core of Rockfill Dams: An Experimental Research Using Sand-Bentonite Mixtures

Ricardo Monteiro-Alves [1,*], Rafael Moran [1,2], Miguel Á. Toledo [1] and Javier Peraita [1]

1   Civil Engineering Department: Hydraulics, Energy, and Environment, E.T.S. de Ingenieros de Caminos, Canales y Puertos, Universidad Politécnica de Madrid, 28040 Madrid, Spain
2   International Centre for Numerical Methods in Engineering, Universitat Politècnica de Catalunya, Campus Norte, 08034 Barcelona, Spain
*   Correspondence: ricardo.monteiro@upm.es

**Abstract:** This article presents experimental research focusing on the structural failure of the central core of a rockfill dam using sand-bentonite mixtures. It comprised an extensive geotechnical characterization of soil materials and mixtures, including compaction and strength tests, as well as the construction of 1 m high and 1.5 m wide physical models. The displacements of the cohesive cores were recorded using a tailored measuring system, based on a laser pointer and a mirror, designed to amplify the real displacements. The cohesive cores were extremely sensitive to small oscillations and behaved as rigid bodies, similar to concrete slabs with three fixed sides and another free. The shape and dimensions of the breach formed on the cohesive cores had roughly the same shape and dimensions as the unprotected area. This experimental research has the potential to be used as validation tool for several models available in the literature to predict the failure of embankment dams.

**Keywords:** cohesive core; dam breach; dam failure; dam safety; floods; overflow; overtopping; rockfill dam

## 1. Introduction

Man-made rockfill structures include levees, dikes, and dams built to meet different human needs, as well as embankment-like deposits of homogeneous coarse rockfill usually produced by mining activities, also known as rock drains [1]. Rockfill structures can also be formed by natural processes and the most common are moraine dams [2] and landslide/avalanche dams [3,4].

Overtopping and pipping are the most common causes of failure of rockfill dams [5]. During extreme events such as these, catastrophic failure of rockfill structures comprises two stages: (i) failure of the downstream shoulder and (ii) failure of the impervious element. Both overtopping and piping lead to the formation of a seepage profile at the base of the dam [6–18], which emerges from the downstream toe [1,17,19–22]. Here, the hydraulic gradients and seepage forces are maximum, making this section of the dam prone to failure [1,19]. Therefore, as a consequence, failure starts at the toe for a discharge that must overcome a given threshold [3,23–26] and evolves upstream until it reaches the crest of the dam if the unit discharges are high enough [27]. At this stage, the breach formed on the downstream shoulder exposes a part of the impervious element of the dam, typically a cohesive core or an upstream concrete face, which from this moment is working as a structural element without the support of the rockfill shoulder. Ultimately, the failure of the impervious element is what controls the outflow hydrograph. Extremely wide central cores could probably be autostable to external forces, so in these cases, failure will first be controlled by erosion processes, leading eventually to a later structural collapse. However, when a core is not autostable, structural failure will be the controlling failure mechanism [28,29].

Some mechanical models have been developed to access the stability of central cohesive cores based on the geomechanical strength of the cohesive soil materials and the geometry of the exposed parts of this element for different modes of failure, such as sliding, overturning and bending [30–33]. The computer program RoDaB [34] developed to predict the failure of rockfill dams, can in theory only be used for landslide of avalanche dams, as it was developed based on rockfill physical models without impervious element. There are other computer programs to simulate the failure of embankment dams such as WinDAM C [35], DL Breach [36,37], EMBREA [38,39] (the HR BREACH successor), etc., which are typically applied to simulate the failure of fine homogeneous dams [40–49]. DL Breach and EMBREA were developed with the aim of also applying them in composite dams, but very few laboratory tests have been performed to validate them [50]. These models also include in their codes simplified mechanical models based on the balance of forces to assess the stability to sliding of the core or of a portion of the embankment remaining from the development of processes such as surface or headcut erosion.

The work presented in this paper is part of a wider research work that focused on the failure of rockfill dams with a central core [32]. It first focused on the failure of the downstream slope [11,27], for then focusing on the failure of the cohesive core. This paper presents part of the work developed to study the failure of the cohesive central core, by means of an experimental set of flume tests developed to assess the structural stability of cohesive central cores of rockfill dams, as well as the mechanisms that lead to the catastrophic failure of this type of impervious element. This experimental research has the potential to be used as validation tool for all these models.

## 2. Methodology

### 2.1. Facilities and Instrumentation

This experimental research was conducted at the Hydraulics Laboratory of the *E.T.S. de Ingenieros de Caminos, Canales y Puertos* of the *Universidad Politécnica de Madrid* (UPM) in Spain. The tests were carried out in a straight U-shaped flume with rectangular section and horizontal bottom, 13.7 m long, 2.5 m wide and 1.3 m high (inner dimensions), with an inspection window 4.6 m long and 1.1 m high placed on the left wall (Figure 1a). It should be noted that the physical models were 1.5 m wide, so a longitudinal wall had to be constructed inside the flume 1.5 m from the left wall of the flume (Figure 1b).

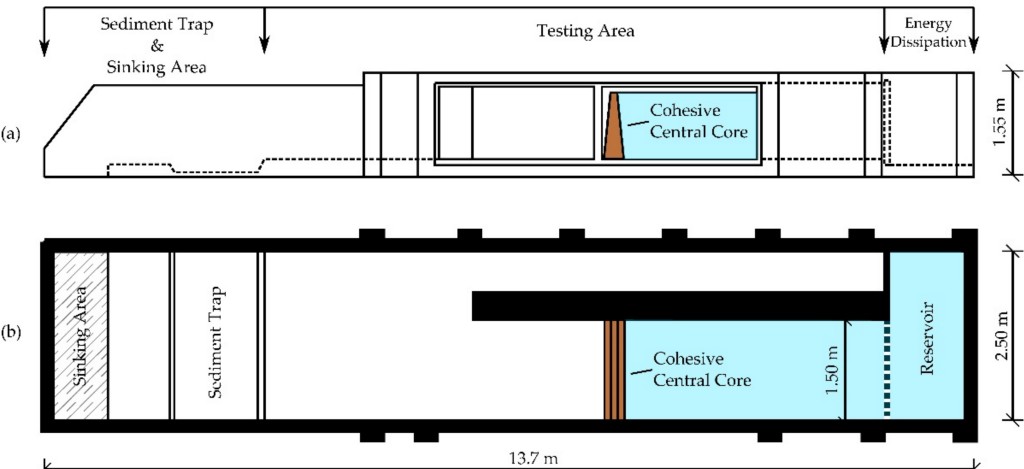

**Figure 1.** Scheme of the location of a cohesive central core inside the UPM flume: (**a**) Side view of the left wall; (**b**) Top view of the flume.

This flume was supplied using an elevated constant water level tank in which the water level was kept constant using a submerged pump with variable frequency drive located inside the underground main tank (270 m³ of capacity through an area of 180 m²). The elevated constant water level tank is connected to the flume through a 0.3 m diameter

pipe with a manual/automatic valve. With the valve fully opened, this system could supply approximately 0.080 $m^3 \cdot s^{-1}$. The reservoir level upstream of the physical models was measured with a P8000 ultrasonic sensor with digital display (Dr. D. Wehrhahn, Hannover, Germany) measuring between 0.07 and 2 m with an accuracy of $\pm 0.0001$ m, located at an upstream distance greater than 2 m. The water level was obtained automatically using National Instruments hardware: an NI-9203 C Series Current Input Module (8-channels, 16-Bit Analog Input, $\pm 20$ mA) assembled in a cDAQ-9172 compact DAQ Chassis (8-Slot, 24-Bit).

The materials used in the construction of the physical models will be thoroughly described in the following sections, but we can advance that these models were constructed with synthetic sand-bentonite mixtures. Small field instrumentation tools were used to control soil materials in different stages of the construction of the physical models.

The SM150T moisture sensor and the HH150 meter from the Delta-T Devices Ltd. (Burwell, Cambridge, UK) were used to measure the water content of sand and bentonite before mixing (the meter has a $\pm 7.5$ mV accuracy with negligible effect on SM150 accuracy and a 0.1% resolution of volumetric reading or 1 mV). Soil materials were weighted using a JWA-30K scale from Jadever Scale Co., Ltd., Taipei, Taiwan, China (measuring range of 0.02 kg to 30 kg with a resolution of 0.001 kg) and mixed using an Umacon UL-190 mixer (0.190 $m^3$ capacity, single-phase electric motor 0.74 kW, 220 V, 50 Hz).

The Geotester Pocket Penetrometer kit and the Humboldt H-4212MH Pocket Shear Vane Tester kit (Ibertest, Madrid, Spain) were used to control the degree of compaction of the physical models. The penetrometer had five plungers 10, 15, 20, and 25 mm in diameter (accuracy $\pm 1$% full scale at a temperature of 20 °C) and the shear vane tester had three vanes with different sizes.

### 2.2. Materials and Geotechnical Characterization

### 2.2.1. Raw Materials

Physical modeling was carried out using an artificial cohesive soil mixture resulting from the mix of a granular material, mainly formed by sand-sized particles, with sodic bentonite clay.

The granular material, gradings of which can be consulted in Table 1, had average sizes of D10 = 0.17 mm, D50 = 0.74 mm, and D60 = 0.94 mm, an average coefficient of uniformity $C_u$ = 5.7, and an average fine content of fines of 5.3% (percentage of material passing the 63 μm sieve). Average values were calculated from gradings obtained for three samples, one following the UNE Standard 103101 (Sieving Method only) and two following the EN ISO Standard 17892-4:2016 (Sieving and Sedimentation Method).

**Table 1.** Gradings of the sandy granular material.

| Sample 1 | | Sample 2 | | Sample 3 | |
|---|---|---|---|---|---|
| Size (mm) | Passing (%) | Size (mm) | Passing (%) | Size (mm) | Passing (%) |
| 12.50 | 100.00 | 6.3 | 100 | 6.3 | 100 |
| 10.00 | 100.00 | 5 | 99.9 | 5 | 99.8 |
| 5.00 | 99.00 | 2 | 89.7 | 2 | 87.3 |
| 2.00 | 85.00 | 1 | 69.9 | 1 | 61.3 |
| 0.40 | 21.00 | 0.63 | 49.7 | 0.63 | 42.3 |
| 0.08 | 2.30 | 0.4 | 31.4 | 0.4 | 25 |
| | | 0.2 | 13.7 | 0.2 | 9.6 |
| | | 0.125 | 8.7 | 0.125 | 5.9 |
| | | 0.1047 | 6.9 | 0.1053 | 4.7 |
| | | 0.08 | 6.7 | 0.08 | 4.6 |
| | | 0.063 | 6.2 | 0.063 | 4.3 |
| | | 0.08 | 5.2 | 0.08 | 3.8 |
| | | 0.0625 | 4.2 | 0.0627 | 2.9 |

**Table 1.** *Cont.*

| Sample 1 | | Sample 2 | | Sample 3 | |
|---|---|---|---|---|---|
| Size (mm) | Passing (%) | Size (mm) | Passing (%) | Size (mm) | Passing (%) |
| | | 0.0551 | 2.7 | 0.0552 | 2.0 |
| | | 0.0395 | 1.5 | 0.0393 | 1.5 |
| | | 0.028 | 1.1 | 0.0281 | 0.9 |
| | | 0.0198 | 1.0 | 0.02 | 0.5 |
| | | 0.0145 | 0.9 | 0.0147 | 0.00 |
| | | 0.0103 | 0.6 | | |
| | | 0.0073 | 0.5 | | |
| | | 0.0052 | 0.5 | | |
| | | 0.0036 | 0.4 | | |
| | | 0.0026 | 0.3 | | |
| | | 0.0015 | 0.1 | | |

The sodic bentonite ('Bentonil C-2'), supplied by Süd-Chemie, has a 6% loss on ignition, a natural moisture of 14%, water absorption of 30 mL/2 g, gradings +100 ASTM sieve under 5% and between 100/200 ASTM sieves under 50%. These characteristics, as well as the chemical properties detailed in Table 2, were provided by the supplier. The final mixture used in the construction of the physical models had a proportion of sand and bentonite ($p_{S:B}$) of 4.56, and water and bentonite contents of $\omega$ = 20% and CB = 18%, respectively. The $p_{S:B}$ ratio, expressed by Equation (1), is the ratio of the dry mass of sand ($W_{S,d}$) to the dry mass of bentonite ($W_{B,d}$).

$$p_{S:B} = W_{S,d}/W_{B,d} \tag{1}$$

**Table 2.** Chemical components of the sodic bentonite.

| Chemical Component | Percentage |
|---|---|
| $SiO_2$ | 64.2% |
| $Al_2O_3$ | 12.1% |
| $Fe_2O_3$ | 2.6% |
| $TiO_2$ | 0.5% |
| $MgO$ | 8.5% |
| $CaO$ | 1.2% |
| $Na_2O$ | 2.3% |
| $K_2O$ | 0.8% |

2.2.2. Soil Mixtures

The final mixture was not known in advance, so a series of mixtures had to be performed previously to define a range of bentonite contents and moistures that could be replicated on a large scale at the Hydraulics Laboratory for the construction of the physical models. A total of 17 sand-bentonite mixtures were prepared with two bentonite contents, 18% and 31%, and moisture contents ranging from 2 to 50%.

2.2.3. Testing

All mixtures were compacted using the standard Proctor procedure (the soil mixtures once prepared were protected with plastic and placed in a humid chamber for 24 h). Indicative soil strength was measured on one or both faces of the compacted standard Proctor samples using the penetrometer and shear vane tester presented in Section 2.1. The compaction tests allowed one to narrow the range of possible mixtures reproducible on a large scale, so, for those within this range, their strength was obtained with Simple Compression tests (UNE 103400:1993) and UU Direct Shear tests (UNE 103401:1998). The relation between mixtures and geotechnical laboratory tests is summarized in Table 3.

**Table 3.** Summary of the geotechnical laboratory tests performed on each mixture.

| Mixture | CB (%) | $\omega$ (%) | Std. Proctor | Simple Compression | Direct Shear |
|---------|--------|--------------|--------------|--------------------|--------------|
| CB18-P1 | 18 | 2.5 | Yes | | |
| CB18-P2 | 18 | 49.7 | Yes | | |
| CB18-P3 | 18 | 41.8 | Yes | | |
| CB18-P4 | 18 | 22.4 | Yes | Yes | Yes |
| CB18-P5 | 18 | 12.4 | Yes | | |
| CB18-P6 | 18 | 28.5 | Yes | | |
| CB18-P7 | 18 | 26.4 | Yes | | |
| CB18-P8 * | 18 | 19.4 | Yes | Yes | Yes |
| CB18-P9 | 18 | 20.4 | Yes | Yes | Yes |
| CB18-P10 | 18 | 21.1 | Yes | Yes | Yes |
| CB31-P1 | 31 | 44.8 | Yes | | |
| CB31-P2 | 31 | 22.5 | Yes | | |
| CB31-P3 | 31 | 14.6 | Yes | | |
| CB31-P4 | 31 | 31.1 | Yes | | |
| CB31-P5 | 31 | 34.1 | Yes | | |
| CB31-P6 | 31 | 36.3 | Yes | | |
| CB31-P7 | 31 | 3.9 | Yes | | |

Note(s): * Mixture used for the short and long-term strength (Simple Compression tests).

The soil samples used in the Simple Compression tests were compacted with the Harvard apparatus, colloquially known as the Mini-Proctor, using the standard Proctor energy ($0.583 \text{ J·cm}^{-3}$). These were cylindrical samples 0.076 m high and 0.038 m in diameter ($86.19 \text{ cm}^3$ mold), compacted in three layers with a 0.5 kg 'Army' hammer and sixteen compaction blows per layer (0.2 m free-fall height).

According to the standard used, the samples should be tested using a deformation velocity ranging from 1 to 2% of the sample height per minute, so the testing velocities should range from $1.27 \times 10^{-5}$ to $2.53 \times 10^{-5} \text{ m·s}^{-1}$. At the Geotechnics Laboratory, the machine allowed only three velocities, 0.02"/min ($8.47 \times 10^{-6} \text{ m·s}^{-1}$), 0.04"/min ($1.69 \times 10^{-5} \text{ m·s}^{-1}$), and 0.06"/min ($2.54 \times 10^{-5} \text{ m·s}^{-1}$), so it was decided to use the intermediate velocity. The apparatus ring constant was 0.286.

Two sets of tests were performed, one to obtain the short-term strength of the soil samples (Phase I), while the other was intended to assess their long-term evolution (Phase II). In any case, the soil samples were compacted using soil mixtures that matured for 24 h in the humid chamber. For Phase I, the compacted samples were tested after 24 h in the humid chamber, and for Phase II, they were tested after 7 and 28 days. Three samples were prepared for every soil mixture tested. The long-term strength set of tests used only one soil mixture (CB18-P8).

Soil samples used in the Direct Shear tests were compacted directly in the upper half of the testing apparatus in three layers with the standard Proctor energy ($0.583 \text{ J·cm}^{-3}$), using the 0.5 kg 'Army' hammer and applying eighteen compaction blows per layer (0.2 m free-fall height). These samples were cylindrical, 0.050 m in diameter and 0.025 m high.

Although the standard stipulates the removal of soil particles greater than 1/10 of the sample height (0.0025 m), it was decided not to remove them before the preparation of the soil mixtures. These were compacted after 24 h in the humid chamber.

For UU tests, the deformation velocity should be high enough to avoid drainage of the pore pressures. According to the standard, it should range from $8.33 \times 10^{-6} \text{ m·s}^{-1}$ (0.5 mm/min) and $2.50 \times 10^{-5} \text{ m·s}^{-1}$ (1.5 mm/min), so it was decided to apply a deformation velocity of $1.67 \times 10^{-6} \text{ m·s}^{-1}$ (1 mm/min). Long-term strength was not assessed with these Direct Shear tests.

### 2.3. Onsite Soil Mix Procedure

Because we were dealing with synthetic mixtures that depend strongly on the bentonite and moisture contents, a detailed procedure had to be defined to pursue reproducibility in the Hydraulics Laboratory of the mixtures compacted and tested at the Geotechnics

Laboratory. In the end, the onsite soil mix procedure was an adaptation to the practice of a basic theoretical approach. Both theoretical and practical approaches are detailed in the following two sections.

### 2.3.1. Theoretical Approach

The first step is to estimate the initial moisture content of a sand-bentonite mixture ($\omega_{SB,i}$) resulting from mixing the sand and the bentonite with their natural moisture contents in a given proportion ($p_{S:B}$). By definition, moisture content ($\omega$) is the ratio of the mass of water to the mass of dry soil. Additionally, by definition, it is known that the dry mass of a generic soil ($W_d$) relates to its apparent mass ($W$), or moistened weight, and its moisture content through Equation (2).

$$W_d = W/(1 + \omega) \tag{2}$$

Given the moisture contents of the sand ($\omega_S$) and bentonite ($\omega_B$), Equation (1) can be rewritten into Equation (3), where $W_S$ and $W_B$ are the apparent weights or moistened weights, of sand and bentonite, respectively.

$$p_{S:B} = [W_S \cdot (1 + \omega_B)]/[W_B \cdot (1 + \omega_S)] \tag{3}$$

So, if the moisture contents of sand and bentonite are known, for a given proportion $p_{S:B}$ we can estimate the apparent weight of one of these two raw materials to add to a given mass of the other. Equation (4) expresses the natural moisture of the generic sand-bentonite mixture that results from mixing the sand and the bentonite with their natural moisture contents.

$$\omega_{SB,i} = [W_{S,d} \cdot \omega_S + W_{B,d} \cdot \omega_B]/[W_{S,d} + W_{B,d}] \tag{4}$$

By combining Equations (4) and (1) we obtain Equation (5).

$$\omega_{SB,i} = [p_{S:B} \cdot \omega_S + \omega_B]/[1 + p_{S:B}] \tag{5}$$

The second and final step is to estimate the amount of water to add to this initial mixture, in order to get the desired final mixture with given bentonite and moisture contents. We are assuming that the mixture after the first step is in a dryer state than the final desired state. The lacking moisture content ($\omega_{add}$) is the difference between the desired moisture ($\omega_{goal}$) and the initial moisture content ($\omega_{SB,i}$). Therefore, the mass of water to add ($W_{H2O,add}$) to the initial mixture can be estimated using Equation (6).

$$W_{H2O,add} = \omega_{add} \cdot (W_{S,d} + W_{B,d}) \tag{6}$$

Combining Equations (1), (3) and (6) we obtain Equation (7), which expresses the mass of water to add to the initial mixture as a function of the apparent (moistured) weight of bentonite, its moisture content, and the desired proportion of sand and bentonite.

$$W_{H2O,add} = \omega_{add} \cdot W_B \cdot (1 + p_{S:B})/(1 + \omega_B) \tag{7}$$

### 2.3.2. Practical Approach

Before the definition of the practical mixing procedure, the SM150T moisture sensor and the HH150 meter had to be calibrated, that is, the moisture measurements obtained with these devices were compared with the UNE Standard 103300:1993 for the determination of moisture content employing the oven-dried methodology.

This device has five modes—Mineral, Peat Mix, Coir, Mineral Wool, and Perlite—so for each one, a series of measurements were performed separately on the sand and bentonite. The average values were then compared to the oven-dried moisture. The UNE standard was applied to a sample of sand (0.546 kg) and another of bentonite (0.269 kg), resulting in moisture contents of 7.1% and 13.8%, respectively.

For sand, the HH150 meter mode that resulted in the smaller error was the Mineral mode, measuring a moisture content of 5.8% ± 1.0% (one standard deviation) for six measurements, so the absolute error was 1.3%. On the other hand, for bentonite, the best mode was the Coir mode resulting in an average moisture content of 16.2% ± 0.7% (one standard deviation) also for six measurements, so the absolute error was 2.4%.

Once the ideal HH150 meter modes were defined, for a given proportion of sand/bentonite ($p_{S:B}$) and moisture content ($\omega_{goal}$), the mixing procedure applied at the Hydraulics Laboratory for the construction of the physical models was as follows:

- Fill a bucket with sand in its initial state (moistured) and weigh it;
- Perform ten measurements with the SM150T moisture sensor and the HH150 meter (Mineral Mode). Use the average value as representative moisture;
- Place this sand inside the concrete mixer and again weigh the bucket to calculate the exact amount of sand used;
- Perform the previous steps until the total mass ranges from 90 to 110 kg;
- Average the moisture content of each bucket and use this value as a representative of the total amount of sand mass placed inside the concrete mixer;
- Open a bentonite bag and perform ten measurements with the SM150T moisture sensor and the HH150 meter (Coir Mode). Use the average value as representative moisture content;
- Use Equation (3) to calculate the mass of bentonite in its initial state to add to the mixture. Place the correct amount of bentonite inside the concrete mixer;
- Start the concrete mixer and let the sand and bentonite mix with the initial moisture contents;
- Use Equation (5) to calculate the initial moisture content of a sand-bentonite mixture, and Equation (7) to calculate the mass of water to add to obtain the desired moisture.
- Add water using a sprinkler while the concrete mixer is working (Figure 2a). As the moisture content increases, the sand-bentonite mixture tends to stick to the mixer walls. In these situations, stop the mixer and use a shovel to remove the mixture from the walls. Restart and stop the number of times necessary to finish adding the total amount of water.
- Once finished, dump the mixture into a trolley and finalize the moisture homogenization with a hoe (Figure 2b).

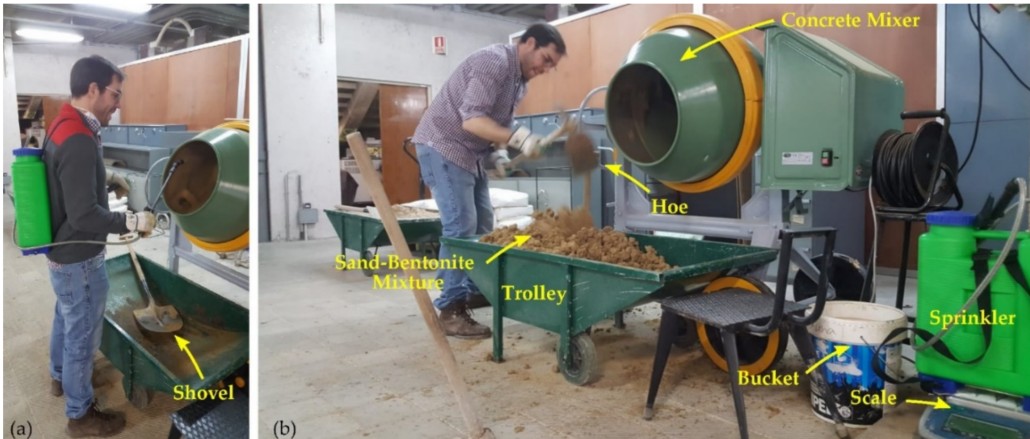

**Figure 2.** Images of the mixing process: (**a**) Sprinkling the sand-bentonite mixture with water while the concrete mixer is working; (**b**) Final moisture homogenization using a hoe.

## 2.4. Construction of the Physical Models

The construction of the cohesive central cores was as follows:

- Drawing of the cohesive core cross-section on both flume walls;

- Once the cross-section was drawn, steel L profiles were stuck to the walls using silicone. L profiles were placed in such a way that one of the faces was in contact with the flume wall and the other with the cohesive core. The main edge should be aligned with the drawn cross-section. The inner part of the L profiles faced the cohesive core (Figure 3a), so in the end, the visible faces were used as a guide for cutting the excess material (Figure 3b).

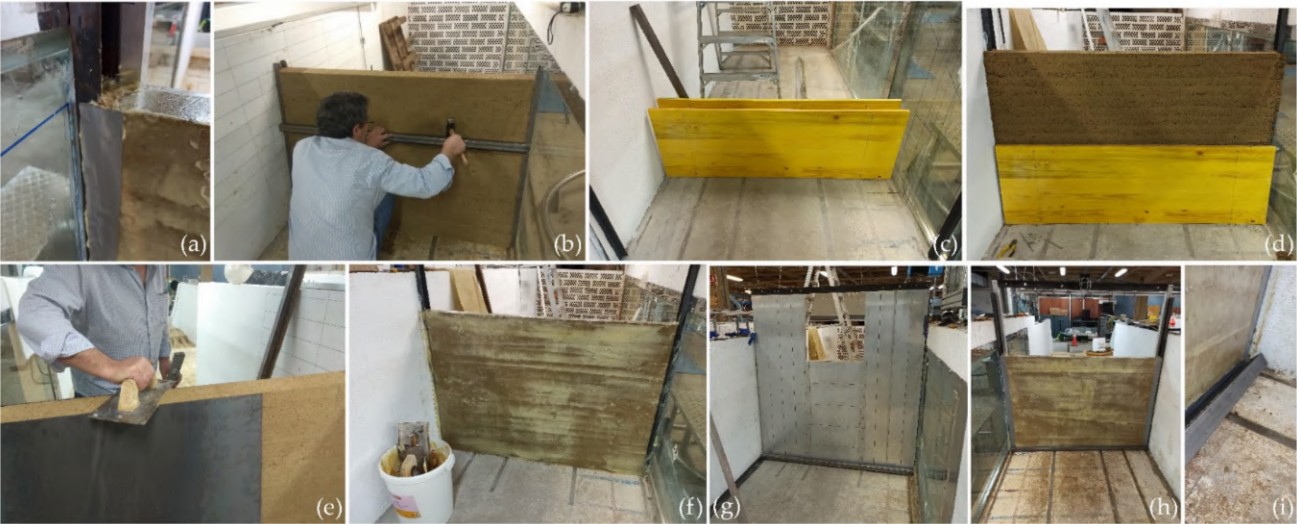

**Figure 3.** Images of the construction of the cohesive central core: (**a**) Placement of tape on the 'vertical' joints between the cohesive core and the metallic L profiles to avoid seepage; (**b**) Cut excess material from the core downstream face; (**c**) Formwork to compact the cohesive material; (**d**) Removal of the two upper panels of the formwork after compaction is finished; (**e**) Cut excess material from the core crest; (**f**) Protection of the downstream face and crest of the core with industrial vaseline (B-2 from Tecmasol); (**g**) Placement of downstream protection to simulate the downstream shoulder support; (**h**) Protection of the upstream face of the core with industrial vaseline (B-2 from Tecmasol); (**i**) Placement of tape on the horizontal joint between the cohesive core and the floor and placement of a metallic profile to avoid lifting and buoyancy of the tape.

- Placement of the formwork to compact the cohesive soil. Two parallel wooden panels 1.49 m long and 0.5 m wide (Figure 3c,d) were placed outside the metallic L profiles and adjusted as much as possible considering the dimensions of the compaction hammer. These two parallel panels were connected by four 5 mm steel threaded bars located near the corners. The distance between panels was fixed using nuts threaded in the 5 mm bars, one inside and two outside the formwork. This prevented both panels from moving during compaction.
- The cohesive core was compacted by layers with depths ranging from 0.05 m to 0.10 m before the compaction. The compaction hammer was an iron prism with two parallel square faces and four rectangular faces 0.095 m long and 0.05 m wide. In one of the four rectangular faces a corrugated steel bar 0.015 m in diameter and 0.77 m long was welded for handling purposes. Compaction was performed by lifting the compaction hammer by approximately 0.2 m and letting it fall freely the number of times needed to reach the desired density. Density was controlled using the Geotester Pocket Penetrometer and the Humboldt H-4212MH Pocket Shear Vane Tester.
- Once the crest elevation was reached, the excess material from the crest was cut with a rectangular trowel using the formwork as reference (Figure 3e). Then, remove the formwork and the 5 mm steel threaded bars from inside the cohesive core. The hole formed by removing these bars was refilled with the cohesive material. The excess material from the downstream face was cut with an artisanal blade roughly as wide as the flume using the previously installed metallic L profiles as a guide (Figure 3b).

- After cutting the excess material, the crest and downstream face were protected with an industrial vaseline (B-2 from Tecmasol) using a rectangular trowel (Figure 3f). This protective grease had a double purpose: to avoid drying of the cohesive material, and to avoid disintegration/precipitation when in contact with water (this phenomenon was observed in a previous trial).

The test procedure and facility were designed to simulate the expected failure process of highly permeable rockfill dams with a cohesive central core. The first damage to the dam starts on the toe when a given overflow threshold is reached. If the overflow continues to increase, the failure will progress upwards until it finally reaches the crest of the dam. From this moment on, the rockfill will be subjected to a plunge jet overflowing the crest of the cohesive core that could lead to the lowering of its elevation, exposing more and more the impervious element [27].

Therefore, placement of the aluminum protection system was intended to simulate the downstream shoulder (Figures 3g and 4). This system was idealized to support the cohesive cores of the reservoir thrust, so clamps were placed on the upper part to hold it tight to the flume walls (Figure 4d), and metallic supports were screwed in the lower part of the walls (Figure 4) to support the toe. To redistribute the concentrated forces of the metallic supports on the toe, a steel U profile (1.5 m long, 0.1 m wide and 0.005 m thick) was placed on the floor of the flume (Figure 4e,f).

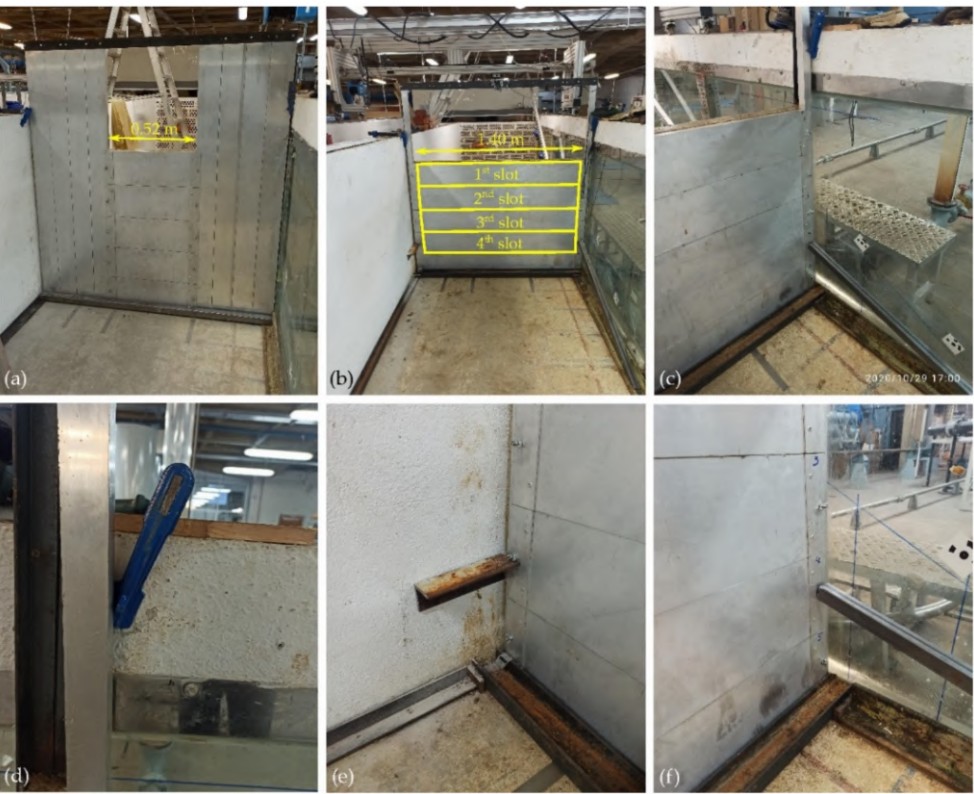

**Figure 4.** Images of the aluminum protection system for the simulation of the support of the core on the downstream shoulder: (**a**) Smaller unprotection width; (**b**) Larger unprotection width; (**c**) Detail of the '1/3' height metallic supports on the left wall of the flume; (**d**) Detail of the clamps used to support the upper section; (**e**) Detail of the metallic supports on the right wall of the flume; (**f**) Detail of the metallic supports on the left wall of the flume.

Two unprotection widths were designed—0.52 m and 1.40 m—and in both cases all horizontal (removable) panels (slots) were rectangular aluminum tubes 0.20 m wide (height) and 0.04 m deep, manufactured with sheets 0.002 m thick. Two T profiles were riveted to both extremes of each horizontal rectangular tube, each T including a hole through

which these were attached to the vertical tubes that had, at different heights, prominent steel threaded bars. The horizontal tubes were held in place using wing nuts. When the wider width of the protection system was tested, it had only one 'vertical' tube (square section 0.04 m wide and 1.6 m long) on each side of the flume. On the other hand, when the narrower unprotection width was tested, four 'vertical' tubes (rectangular section 0.12 m wide, 0.04 m deep, and 1.6 m long) were placed on both sides of the flume.

Both versions of the protection system were reinforced with two steel L profiles (1.5 m long, 0.05 m wide, and 0.005 m thick) placed horizontally at both extremes of the 'vertical' tubes.

### 2.5. Measuring the Displacements of the Cohesive Core

To measure the small displacements of the cohesive core, a system based on a laser pointer and a mirror was designed to amplify the real displacements (Figure 5). The projected laser beam trajectory ranged from the right wall to the left wall of the flume passing over white aluminum bars (rectangular section 0.06 m wide and 0.02 m deep). Two bars were screwed to the walls, and one was placed transversely to the flume at a distance of 2.4 m from the laser pointer.

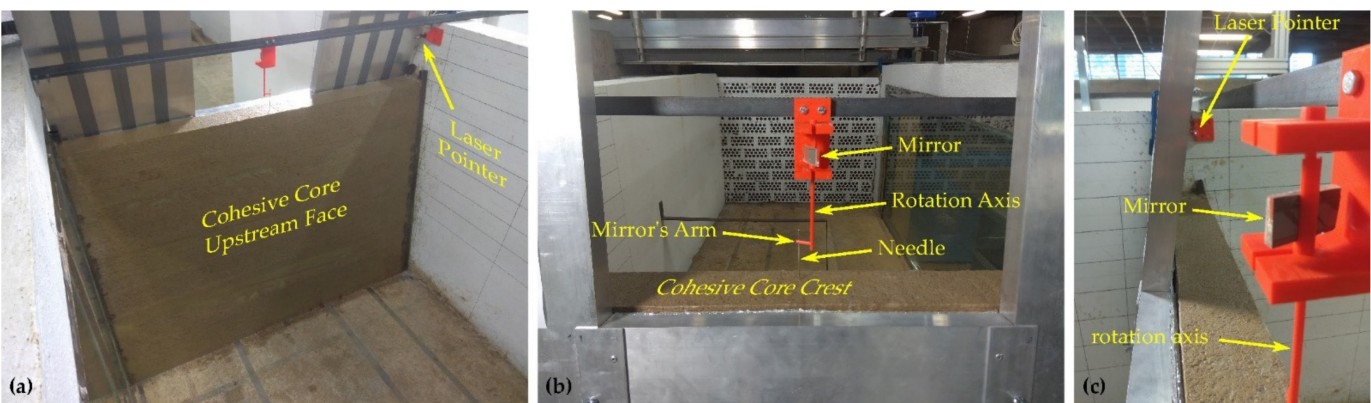

**Figure 5.** Images of the displacement measuring system: (**a**) View from the upstream side; (**b**) View from the downstream side; (**c**) View along the cohesive core crest.

The amplification factors vary as the projected laser beam travels along these three planes (Figure 6). To calculate the amplified movements, we need to define the following variables: the rotation angle ($\gamma$), the real displacements along the longitudinal axis of the flume ($s$), the amplified displacements ($S$), the angle of the original laser beam direction (perpendicular to the flume) with the projected laser beam direction before any rotation ($\beta_0$); the angle of the original laser beam direction (perpendicular to the flume) with the initial position of the mirror's arm on which the needle leans on ($\alpha_0$), the distance along the longitudinal axis of the flume between the initial position of the needle and the laser pointer ($x_0$), the distance along the right wall of the flume between the laser pointer and the initial position of the projected laser beam point ($X_0$), the angle of the original laser beam direction (perpendicular to the flume) with the new position of the rotating arm of the mirror ($\alpha$), and the angle of the original laser beam direction (perpendicular to the flume) with the projected laser beam direction after rotation ($\beta$).

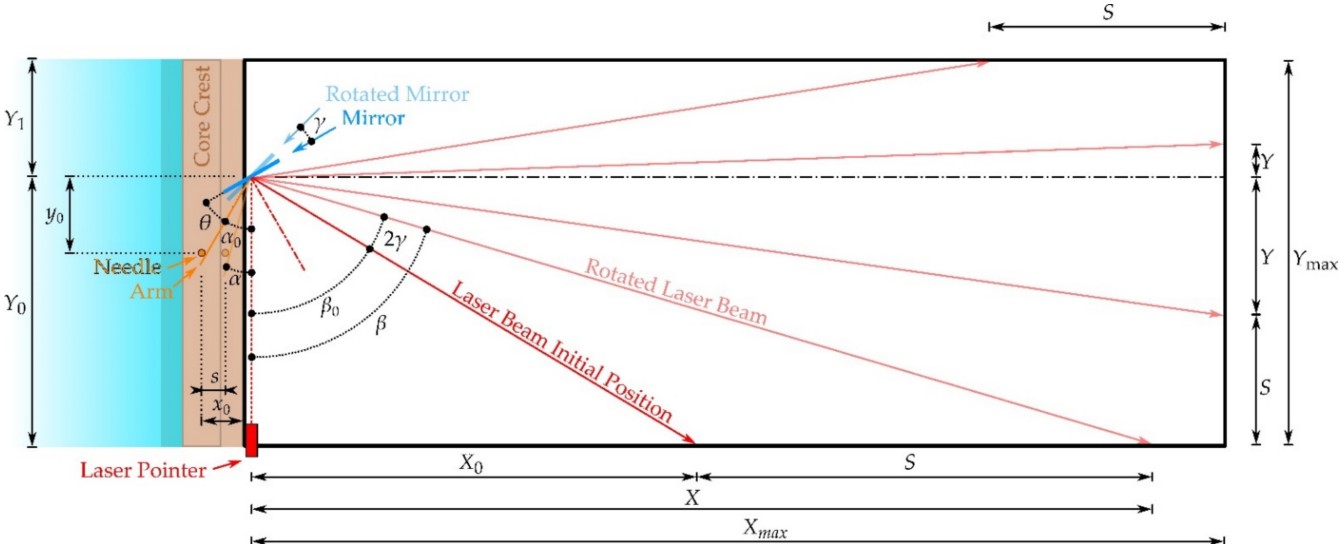

**Figure 6.** Top view scheme of the measuring system designed to control the displacements of the cohesive core. The dimensions of the elements of this system are not scaled to their true dimensions.

Angles can be defined using Equations (8)–(12).

$$\alpha_0 = 90° - \theta - \beta_0/2 \text{ or } \alpha_0 = \tan^{-1}(x_0/y_0) \tag{8}$$

$$\beta_0 = \tan^{-1}(X_0/Y_0) \tag{9}$$

$$\alpha = \tan^{-1}[(x_0 - s)/y_0] \tag{10}$$

$$\beta = \beta_0 + 2\gamma \text{ or } \beta = \tan^{-1}(X/Y_0) \tag{11}$$

$$\gamma = \alpha_0 - \alpha \tag{12}$$

On the right wall of the flume, the amplified displacements of the cohesive core are calculated with Equation (13).

$$S = X - X_0 \tag{13}$$

On the front panel, the calculations will depend on the position of the laser beam with respect to the longitudinal axis on which the mirror is located. If the laser beam projection is to the right of the mirror, the amplified displacement will be $S = Y_0 - Y$, where $Y$ is expressed by Equation (14). On the other hand, if it is projected on the left side, then $S = Y_0 + Y$ with $Y$ expressed by Equation (15).

$$Y = X_{max} \cdot \tan(90° - \beta) \tag{14}$$

$$Y = X_{max} \cdot \tan(\beta - 90°) \tag{15}$$

When the projection is on the left wall of the flume, the amplified displacements are calculated with Equation (16).

$$S = X_{max} - Y_1 \cdot \tan(180° - \beta) \tag{16}$$

The displacements were measured in the center of the cohesive core (longitudinal axis of the flume), and the adopted values for the initial state variables were: $X_0 = 0.4$ m, $Y_0 = 0.77$ m, $Y_1 = 0.71$ m, $y_0 = 0.2$ m, and $\theta = 0°$.

*2.6. Testing Procedure*

Taking into account the failure process of rockfill dams with cohesive central core described in the last paragraphs of Section 2.4, the tests were generally performed by maintaining a constant hydraulic load (constant reservoir level coincident with the elevation

of the crest to avoid interference with the displacement measuring system) and testing different degrees of unprotection by removing horizontal slots (rectangular tubes 0.20 m wide) from top to bottom until the cohesive central core failed.

For each degree of unprotection, the displacements were recorded 10, 100, and 1000 s after the removal of a given slot. By analyzing the displacements in the logarithmic scale, if these displacements completely stopped or tended to slow down, then the core was assumed to be stable for this particular degree of unprotection, and so proceeding with the removal of the next slot. On the other hand, if the displacements were observed to continue growing, with constant velocity or accelerating, then the displacements should be recorded after another 1000 s (2000 s accumulated).

Again, if these displacements completely stopped or tended to slow down, then the core was assumed to be stable for this particular degree of unprotection and so proceeding with the removal of the next slot. If the displacements did not stop, this procedure should be continued until reaching an accumulated total time of 4000 s, moment for which a simple calculation should be done to understand how far the cohesive core was from potential failure.

Taking into account the velocity with which the core was moving, we could make an estimation of the time to reach a theoretical displacement of 5% of the width of the breach, assumed to be the maximum displacement without failure. In addition, a factor of safety could be estimated by comparing the maximum displacement with that observed until this moment. If the failure time and the factor of safety were, respectively, greater than 1 year and 2, then it was assumed that the core was stable for this degree of unprotection. On the other hand, that is, if the failure time or the factor of safety were, respectively, lower than 1 year and 2, the displacements should be recorded again after another 4000 s (8000 s accumulated). If in the end, the displacements continue growing, then the core was assumed to fail for this degree of unprotection degree even though it did not break until this moment.

### 2.7. Laboratory Experimental Tests

The Main Laboratory Experiments (MAIN) included three cohesive cores, all symmetric with the same geometry and dimensions: 1 m high, 0.06 m wide crest, and 0.10 m wide base. Two cohesive cores, denoted MAIN1 and MAIN2, were tested with the 0.52 m wide unprotection slots (Figure 4a) and one, MAIN3, with the 1.40 m wide slots (Figure 4b). All three tests were performed following the procedure defined in Section 2.6.

The sand-bentonite mixture, which was intended to be the same for the three tests ($p_{S:B}$ = 4.56, $\omega$ = 20%, and CB = 18%), was prepared separately for the construction of each cohesive core. The Main Laboratory Experiments included also the performance of an extra test using the failed geometry of the cohesive core MAIN3. This test was denoted MAIN3+. The displacements were not measured in this extra test because the measuring system was not prepared for a different core height. Instead of maintaining a constant hydraulic load and testing different degrees of unprotection, we maintained the degree of unprotection constant and incremented the overflow discharge until failure.

Preliminary Laboratory Experiments (PRELIM), which goal was to help defining the test protocol, included two symmetric cohesive cores, PRELIM1 and PRELIM2, tested with the 0.52 m wide unprotection slots (Figure 4a), and without measuring the displacements because they were carried out before the definition of the displacement measuring system. The soil material was the same as in the Main Laboratory Experiments.

Test PRELIM1, constructed from scratch, was 1 m high with a 0.15 m wide crest and a 0.50 m wide base. On the other hand, the geometry of PRELIM2 was obtained by trimming the original PRELIM1 cohesive core, that did not fail. So, PRELIM2 was 0.94 m high with a 0.12 m wide crest and a 0.45 m wide base.

Regarding the hydraulic loading, PRELIM1 was performed with overflow by imposing a constant inflow to the flume of 0.014 $m^3 \cdot s^{-1}$ which resulted in a hydraulic head over the

crest of 0.05 m. On the other hand, PRELIM2 was performed without overflow and kept the reservoir level constant and coincident with the crest elevation.

Table 4 summarizes both preliminary (2 tests) and main (4 tests) sets of tests.

**Table 4.** Laboratory experimental set of tests.

| | | Test | | | | | |
|---|---|---|---|---|---|---|---|
| Parameter | Symbol | PRELIM1 | PRELIM2 | MAIN1 | MAIN2 | MAIN3 | MAIN3+ |
| Core height (m) | $H$ | 1 | 0.94 | 1 | 1 | 1 | $\approx$0.6 |
| Crest width (m) | $l_c$ | 0.15 | 0.12 | 0.06 | 0.06 | 0.06 | $\approx$0.076 |
| Base width (m) | $l_b$ | 0.5 | 0.45 | 0.1 | 0.1 | 0.1 | 0.1 |
| Reservoir level (m) | $H_r$ | $\approx$1.05 | $\approx$0.94 | $\approx$1 | $\approx$1 | $\approx$1 | $\approx$0.68 * |
| Unprotection width (m) | $w'$ | 0.52 | 0.52 | 0.52 | 0.52 | 1.40 | 1.40 |

Note(s): * Reservoir level for which failure occurred.

## 3. Results

### 3.1. Validation of the Onsite Soil Mix Procedure

The mix procedure on site, thoroughly described in Section 2.3, was validated before the construction of the laboratory tests. Two mixtures were prepared in the Hydraulics Laboratory: CB18-P4 ($\omega$ = 22.4%) and CB18-P8 ($\omega$ = 19.4%). Both, once prepared, were placed in a trolley (Figure 2b) and covered with a plastic sheet for approximately 24 h. After this, a sample of each mixture was obtained and tested according to UNE Standard 103300:1993 for the determination of the moisture content by employing the oven dried methodology. The moisture content resulting from these tests was 22.4% and 18.3% for the CB18-P4 and CB18-P8 samples, respectively, so the differences, 0 and 1.1%, are very small taking into account all the sources of error involved in the mixing process.

### 3.2. Compaction Tests

The results of the compaction tests are summarized in Table 5 and Figure 7 These results, which are far from conventional and will be discussed later, are not so uncommon when dealing with bentonite.

**Table 5.** Summary of the standard Proctor Tests.

| Mixture | CB (%) | $\omega$ (%) | $\rho_d$ (kg·m$^{-3}$) | $\rho$ (kg·m$^{-3}$) |
|---|---|---|---|---|
| CB18-P1 * | 18 | 2.5 | 1902 | 1950 |
| CB18-P5 | 18 | 12.4 | 1747 | 1963 |
| CB18-P8 | 18 | 19.4 | 1679 | 2005 |
| CB18-P9 | 18 | 20.4 | 1685 | 2030 |
| CB18-P10 | 18 | 21.1 | 1653 | 2001 |
| CB18-P4 | 18 | 22.4 | 1608 | 1969 |
| CB18-P7 | 18 | 26.4 | 1512 | 1911 |
| CB18-P6 | 18 | 28.5 | 1414 | 1818 |
| CB18-P3 | 18 | 41.8 | 1214 | 1722 |
| CB18-P2 | 18 | 49.7 | 1116 | 1670 |
| CB31-P7 * | 31 | 3.9 | 1795 | 1864 |
| CB31-P3 | 31 | 14.6 | 1597 | 1830 |
| CB31-P2 | 31 | 22.5 | 1586 | 1944 |
| CB31-P4 | 31 | 31.1 | 1406 | 1843 |
| CB31-P5 | 31 | 34.1 | 1345 | 1804 |
| CB31-P6 | 31 | 36.3 | 1290 | 1758 |
| CB31-P1 | 31 | 44.8 | 1165 | 1687 |

Note(s): * These densities are the mean values including the results of the repeated test.

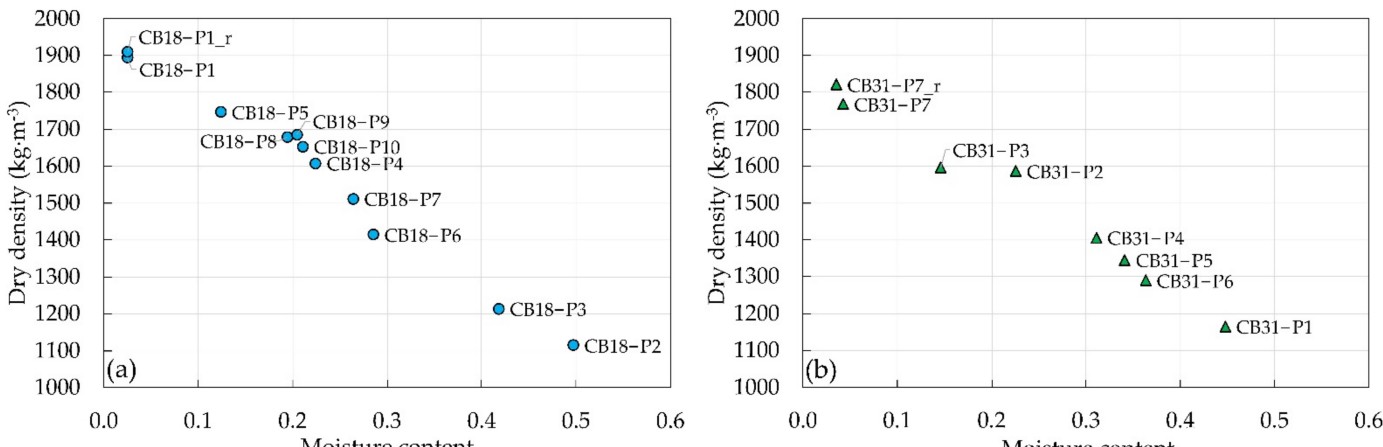

**Figure 7.** Summary of the Standard Proctor Tests: (**a**) Soil mixtures with a bentonite content of 18%; (**b**) Soil mixtures with a bentonite content of 31%. The subscript 'r' stands for 'repeated'.

Figure 8a,b present, respectively, the indicative strength measurements obtained with the Geotester Pocket Penetrometer and the Humboldt H-4212MH Pocket Shear Vane Tester, performing four and three measurements on average on each face of the Proctor samples with each device. The red dots represent an estimate for a CB18 mixture with an optimal moisture content of 20% and a relative maximum dry density $\rho_{d,max}$ = 1692 kN·m$^{-3}$. The Proctor optimal moisture content is discussed in Section 4.1.

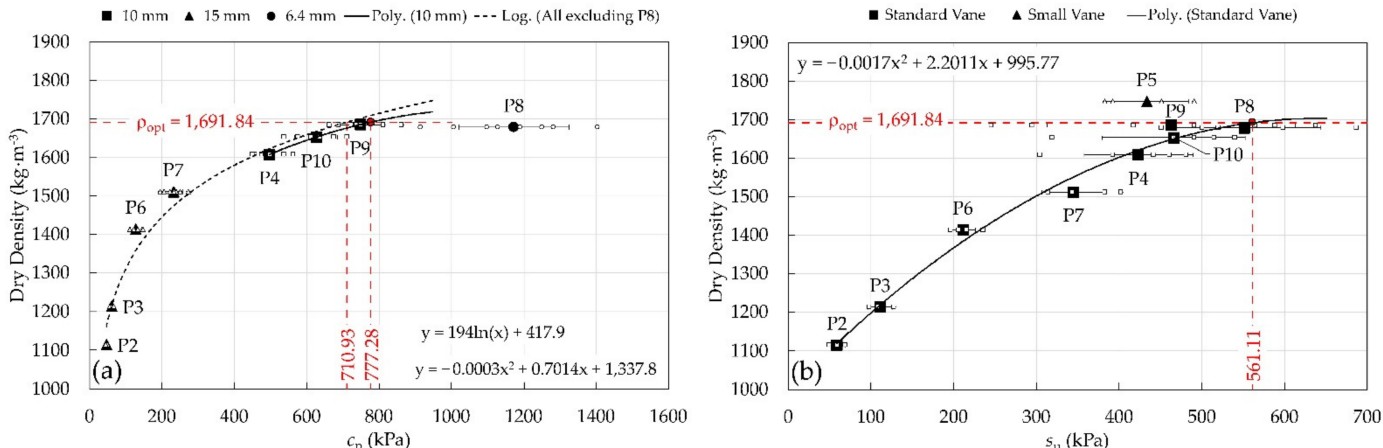

**Figure 8.** (**a**) Strength measurements obtained with the Geotester Pocket Penetrometer distinguished by the size of the plunger. The logarithmic regression curve was fitted to all tests except CB18-P8, and the quadratic regression curve to the tests CB18-P4, CB18-P9 and CB18-P10; (**b**) Measurements made with the Pocket Shear Vane Tester distinguished by the vane size. The quadratic regression curve is fitted using all data points except CB18-P5 as it was obtained with the smaller vane instead of the standard; In both charts, the big solid dots represent the average strength value, the small white dots represent each of the measurements, and the horizontal lines represent the extension of one standard deviation from the mean.

These charts use box plots that identify the mean values, the standard deviation from the mean, and all the measurements. They also identify the size of the plungers and vanes: (i) Geotester Pocket Penetrometer 6.4 mm plunger in CB18-P8; 10 mm plunger in CB18-P4, CB18-P9, and CB18-P10; 15 mm plunger in CB18-P2, CB18-P3, CB18-P6, and CB18-P7; One extra measurement on CB18-P4, and CB18-P10. (ii) Pocket Shear Vane Tester standard vane (medium size) was used on all compaction tests except on CB18-P5, in which it was used the smaller vane.

### 3.3. Undrained Unconfined Shear Strength

All Simple Compression compacted soil samples presented a peak stress without reaching the critical state, that is, deformation for constant stress (Figure A1 in Appendix A). In Phase I, from the stress–strain relationship we obtained an average elastic modulus of $E_{\text{u,Phase I}}$ = 3.9 MPa ± 0.8 MPa (one standard deviation). This modulus increased approximately 59% on average to $E_{\text{u,Phase II}}$ = 6.3 MPa ± 0.8 MPa with Phase II samples. Tables 6 and 7 summarize, respectively, Phase I and Phase II of the Simple Compression tests, detailing the target moisture content ($\omega_{\text{Proctor}}$) for each soil mixture and soil sample, the moisture content of the soil mixture ($\omega_{\text{mix}}$) used in the compaction of the samples, the moisture content of the soil samples ($\omega_{\text{sample}}$) using a fraction of material extracted from the interior of the failed samples, the target apparent or moistened density ($\rho_{\text{Proctor}}$) that each soil sample should have, the apparent (moistened) density of each sample ($\rho_{\text{sample}}$), the unconfined shear strength ($c_{\text{u}}$), calculated as half of the unconfined compressive strength ($q_{\text{u}}$) that in these tests was the peak strength (Equation (17)), and the strain ($\varepsilon_{\text{failure}}$) and time ($t_{\text{failure}}$) for which the peak compressive strength was reached.

$$c_{\text{u}} = 0.5 \cdot (\sigma_1 - \sigma_3) = 0.5 \cdot (q_{\text{u}} - 0) = 0.5 \cdot q_{\text{u}} \tag{17}$$

**Table 6.** Summary of the Simple Compression Tests (Phase I). The subscript 'r' stands for 'repeated' and NA for 'Not Available'.

| Mixture | Sample | $\omega_{\text{Proctor}}$ (%) | $\omega_{\text{mix}}$ (%) | $\omega_{\text{sample}}$ (%) | $\rho_{\text{Proctor}}$ (kg·m$^{-3}$) | $\rho_{\text{sample}}$ (kg·m$^{-3}$) | $c_{\text{u}}$ (kPa) | $\varepsilon_{\text{failure}}$ (%) | $t_{\text{failure}}$ (s) |
|---|---|---|---|---|---|---|---|---|---|
| CB18-P4 | 1<br>2 | 22.4 | 23.3 | 22.3<br>22.4 | 1969 | 1988<br>1967 | 22.13<br>20.38 | 5.33<br>5.33 | 240<br>240 |
| CB18-P4r | 1<br>2 | 22.4 | 22.2 | 18.6<br>19.7 | 1969 | 2050<br>2035 | 30.50<br>28.71 | 4.67<br>6.67 | 210<br>300 |
| CB18-P8 | 1<br>2<br>3 | 19.4 | NA | 20.0<br>19.8<br>19.6 | 2005 | NA<br>2026<br>2015 | 29.33<br>29.91<br>23.30 | 4.67<br>4.67<br>5.33 | 210<br>210<br>240 |
| CB18-P8r | 1<br>2<br>3 | 19.4 | 19.6 | 23.6<br>21.8<br>21.1 | 2005 | 1956<br>1987<br>2002 | 15.74<br>20.30<br>21.66 | 7.33<br>7.33<br>7.33 | 330<br>330<br>330 |
| CB18-P9 | 1<br>2 | 20.4 | 20.4 | 19.5<br>19.4 | 2030 | 2031<br>2029 | 29.65<br>29.32 | 6.00<br>5.33 | 270<br>240 |
| CB18-P10 | 1<br>2 | 21.1 | 20.9 | 20.5<br>20.2 | 2001 | 2031<br>2024 | 26.70<br>25.59 | 7.33<br>5.33 | 330<br>240 |

**Table 7.** Summary of the Simple Compression Tests (Phase II). The subscript 'r' stands for 'repeated' and NA for 'Not Available'.

| Mixture | Sample | $\omega_{\text{Proctor}}$ (%) | $\omega_{\text{mix}}$ (%) | $\omega_{\text{sample}}$ (%) | $\rho_{\text{Proctor}}$ (kg·m$^{-3}$) | $\rho_{\text{sample}}$ (kg·m$^{-3}$) | $c_{\text{u}}$ (kPa) | $\varepsilon_{\text{failure}}$ (%) | $t_{\text{failure}}$ (s) |
|---|---|---|---|---|---|---|---|---|---|
| CB18-P8 (7 days) | 1<br>2<br>3 | 19.4 | 19.7 | 19.2<br>19.6<br>18.6 | 2005 | 2020<br>2015<br>2027 | 39.52<br>37.47<br>41.92 | 2.67<br>3.33<br>2.67 | 120<br>150<br>120 |
| CB18-P8 (28 days) | 1<br>2<br>3 | 19.4 | 20.0 | 18.8<br>18.6<br>18.7 | 2005 | 2041<br>2042<br>2032 | 38.98<br>40.16<br>40.72 | 4.00<br>4.00<br>2.67 | 180<br>180<br>120 |

### 3.4. Undrained Direct Shear Strength

The UNE 103401:1998 standard specifies that from this test we obtain the strength parameters $c_{\text{u}}$ and $\phi_{\text{u}}$. However, to avoid confusion with the unconfined shear strength obtained with the Simple Compression tests (defined in this paper as $c_{\text{u}}$), we decided

to rename the Direct Shear strength parameters to $c_{uu}$ and $\phi_{uu}$. These parameters were obtained for the following soil samples: CB18-P4, CB18-P8, CB18-P9, and CB18-P10.

These tests are summarized in Table 8 that details the target moisture content ($\omega_{Proctor}$) for each soil mixture and soil sample, the initial moisture content of the soil samples ($\omega_{sample}$) using a fraction of the excess material used for compaction, the target apparent or moistened density ($\rho_{Proctor}$) that each soil sample should have, the apparent or moistened density of each soil sample ($\rho_{sample}$), the maximum shear stress ($\tau_{max}$) using the corrected shear surface area according to the UNE 103401:1998 standard formulations, and the strength parameters $c_{uu}$ and $\phi_{uu}$. Only two samples showed a clear stress peak and reached the critical state after that. For the rest of the samples, the stress–strain relationship did not stop increasing or reached the critical state without a clear peak [32] (p. 455–461).

**Table 8.** Summary of the Undrained Direct Shear Tests. The subscript 'r' stands for 'repeated'.

| Mixture | Sample | $\omega_{Proctor}$ (%) | $\omega_{sample}$ (%) | $\rho_{Proctor}$ (kg·m$^{-3}$) | $\rho_{sample}$ (kg·m$^{-3}$) | $\tau_{max}$ (kPa) | $c_{uu}$ (kPa) | $\phi_{uu}$ (°) |
|---|---|---|---|---|---|---|---|---|
| CB18-P4 | 1 | 22.4 | 22.6 | 1969 | 1954 | 42.87 | 42.16 | 16.1 |
| | 2 | | 22.7 | | 1949 | 45.05 | | |
| | 3 | | 25.8 | | 1994 | 31.02 | | |
| | 3r | | 25.8 | | 1947 | 31.54 | | |
| CB18-P8 | 1 | 19.4 | 19.8 | 2005 | 2005 | 32.88 | 30.61 | 50.2 |
| | 2 | | 19.8 | | 2005 | 43.85 | | |
| | 3 | | 19.8 | | 2005 | 54.06 | | |
| CB18-P9 | 1 | 20.4 | 20.5 | 2030 | 2025 | 37.85 | 36.98 | 35.3 |
| | 2 | | 20.7 | | 2030 | 45.59 | | |
| | 3 | | 20.2 | | 2030 | 50.47 | | |
| CB18-P10 | 1 | 21.1 | 21.0 | 2001 | 2002 | 23.33 | 23.36 | 52.8 |
| | 2 | | 21.5 | | 2000 | 42.34 | | |
| | 3 | | 21.5 | | 2000 | 47.22 | | |
| | 3r | | 21.2 | | 2000 | 47.76 | | |

*3.5. Cohesive Central Cores Density as Built*

Figure 9 presents the strength measurements performed during the construction of the cohesive cores with the Geotester Pocket Penetrometer ($c_p$) using the 10 mm plunger and the Humboldt H-4212MH Pocket Shear Vane Tester ($s_u$) using the standard vane. The Penetrometer 15 mm plunger was used only to control the degree of compaction of the third and fourth layers of the first cohesive core, tested in PRELIM1 and PRELIM2 (Figure 9a). This could explain the low resistance to penetration of these layers. This will be discussed later on Section 4.2. Figures from (a) to (d) present, respectively, the control of compaction performed on the cores used for testing PRELIM1 and PRELIM2, MAIN1, MAIN2, and MAIN3 and MAIN3+ (Table 4).

The Pocket Shear Vane Tester was the tool that presented lower variability in the results. Measurements carried out with this tool were also closer to the expected value ($s_{u,goal}$ = 561 kPa for a CB18 mixture with an optimal moisture content of 20.0% and a relative maximum dry density $\rho_{d,max}$ = 1692 kN·m$^{-3}$, as detailed in Figure 8). These measurements were lower than expected, on average 10.5% $\pm$ 6.2% (one standard deviation). The larger and smaller differences occurred for tests PRELIM1 and PRELIM2 (20.9%, Figure 9a) and MAIN2 (4.6%, Figure 9c), respectively.

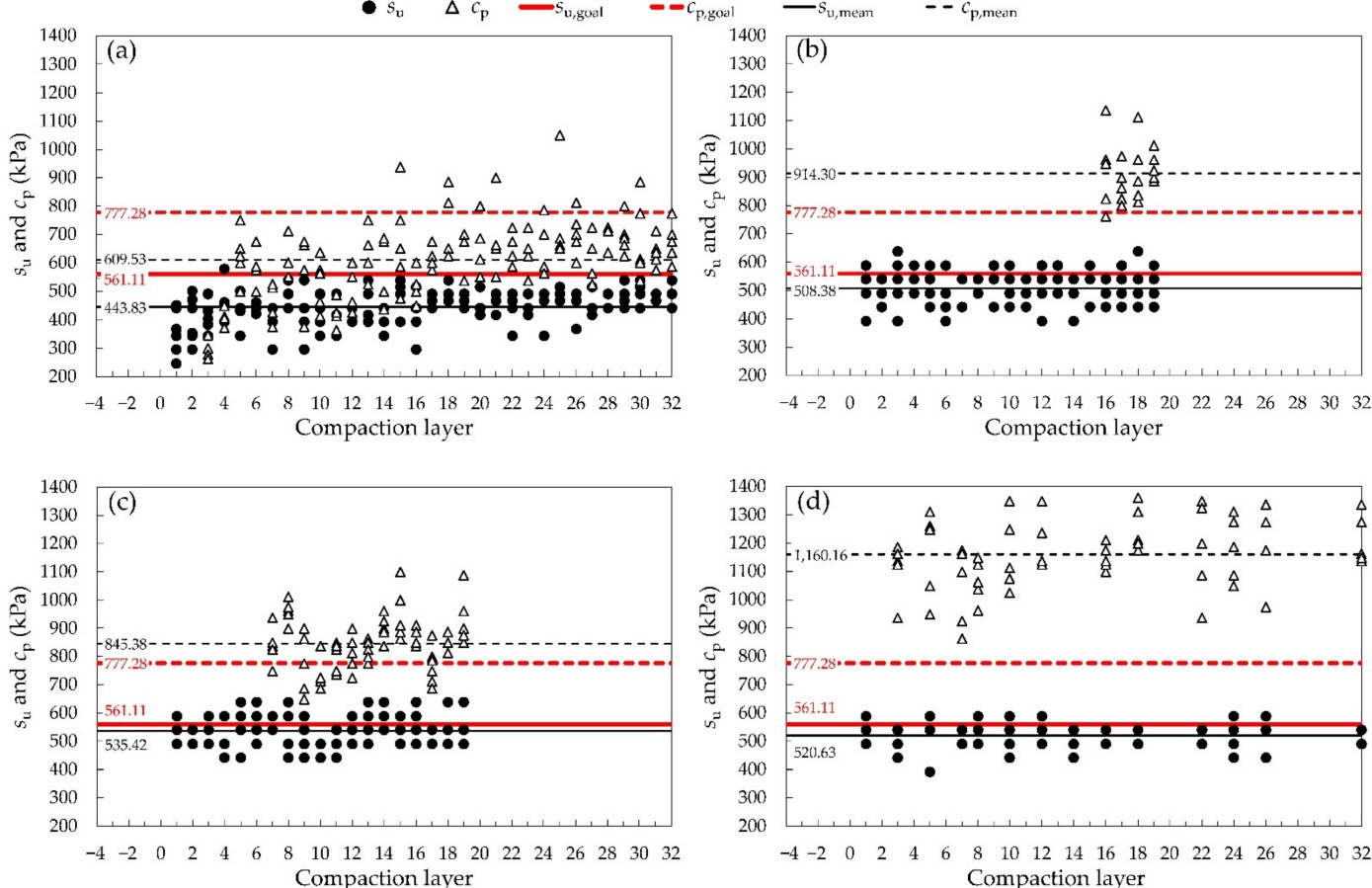

**Figure 9.** Strength measurements obtained during the construction of cohesive cores with the Geotester Pocket Penetrometer ($c_p$) and the Humboldt H−4212MH Pocket Shear Vane Tester ($s_u$). Cohesive cores used for (**a**) PRELIM1 and PRELIM2, (**b**) MAIN1, (**c**) MAIN2; (**d**) MAIN3 and MAIN3+ (Table 4). The strengths $s_{u,goal}$ and $c_{p,goal}$ are estimates for a CB18 mixture with an optimal moisture content of 20.0% and a relative maximum dry density $\rho_{d,max} = 1692$ kN·m$^{-3}$ (Figure 8). On the other hand, $s_{u,mean}$ and $c_{p,mean}$ are the average strengths using all measurements.

### 3.6. The Reservoir Level and the Displacements of the Cohesive Core

The extra reinforcements (Figure 4e,f) applied to the aluminum protection system at approximately one-third of the height of the physical models on tests MAIN3 and MAIN3+ were very effective in controlling the displacements of this protection system. This cohesive core did not experience any displacement during the reservoir filling period and we did not have to reset the displacements to zero before starting to remove the protection slots (Figure 10e). Because these extra reinforcements were not applied to the first two Main Laboratory Experiments, the displacements had to be reset to zero in tests MAIN1 (Figure 10a) and MAIN2 (Figure 10c). The core crest on the first and second tests moved downstream 0.014 m and 0.007 m, respectively.

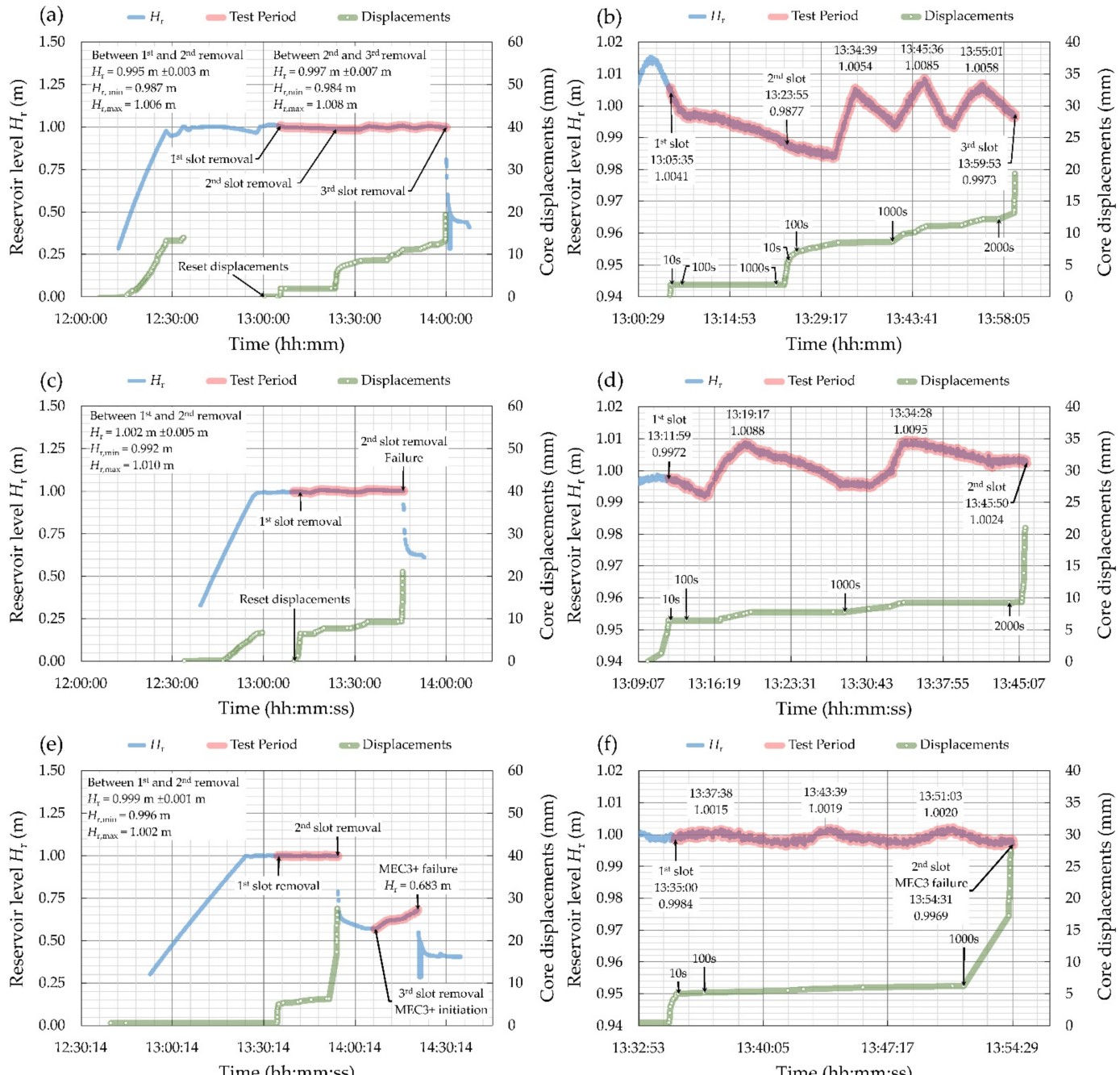

**Figure 10.** Reservoir level elevation and the cohesive core displacements. (**a**) Test MAIN1 including the reservoir filling period; (**b**) Detail of the test MAIN1; (**c**) Test MAIN2 including the reservoir filling period; (**d**) Detail of the test MAIN2; (**e**) Tests MAIN3 and MAIN3+ including the reservoir filling period; (**f**) Detail of the test MAIN3. The times 10 s, 100 s, 1000 s, and 2000 represent the time in seconds elapsed from the moment a given protection slot was removed.

It is remarkable that the cohesive cores were very sensitive to the variations of the reservoir level, even for a rising of a few millimeters. For example, in test MAIN3 (Figure 10f) just after the removal of the first protection slot, was observed a rising of just 0.0031 m resulted in a displacement of 0.80 mm of the crest in the downstream direction (Figure 10f; compare $t = 3309$ s and $t = 3409$ s in Table A3). The cohesive cores were displaced even when the reservoir level was lower than in previous moments. For example, in test MAIN2 (Figure 10d), the displacements start to increase before the second peak is reached (time 13:34:28), but the reservoir levels for which these displacements occur are lower than those

experienced moments before during the first peak (time 13:19:17). Sometimes, the displacements of the cohesive cores in response to a change in hydraulic loading had a little delay. This can be observed in Figure 10d when the reservoir starts to rise after the removal of the first protection slot. The displacements registered during tests MAIN1, MAIN2, and MAIN3 are detailed, respectively in Tables A1–A3 in Appendix B.

### 3.7. The Mechanics of Failure

Concerning the Preliminary Laboratory Experiments, it should be noted that they were used to help defining the protocol to follow during the Main Laboratory Experiments. PRELIM1 did not fail. We removed the first four protection slots without any damage to the cohesive core. The first level of unprotection was maintained for 11 min, the second for 12 min, the third for 13 min, and the fourth for 9 min. Test PRELIM2 had the same outcome, it resisted the removal of all protection slots, even for one more level of unprotection, that is, the removal of the fifth slot.

Regarding the Main Laboratory Experiments, MAIN1, MAIN2, MAIN3, and MAIN3+, all cohesive cores failed in a similar way, as rigid bodies like concrete slabs (Figure 11).

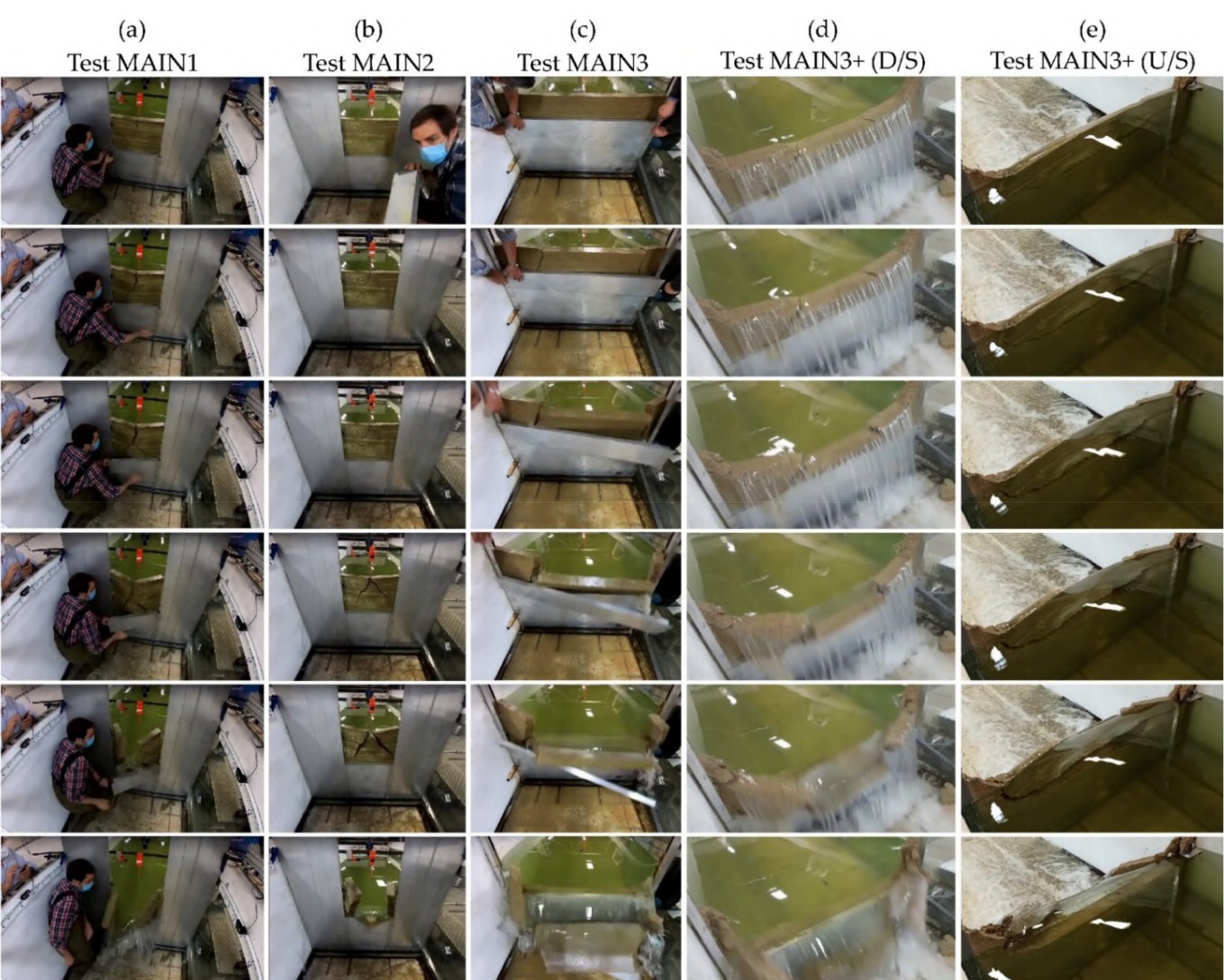

**Figure 11.** Sequential images of the failures of the Main Laboratory Experiments. (**a**) Test MAIN1; (**b**) Test MAIN2; (**c**) Test MAIN3; (**d**) Extra test MAIN3+ from the downstream side; (**e**) Extra test MAIN3+ from the upstream side.

Cracks developed on the downstream face until three blocks were formed, two of them rotating around the 'vertical' axes located on the lateral walls of the breach and one around the 'horizontal' axis located at the base of the breach. These observations are consistent with other experimental research [29], where a thin core with a 0.3 ratio of the width of the cohesive core base to its height also collapsed in the same way.

The first test, MAIN1 ($w' = 0.52$ m), failed when the third protection slot was removed. A vertical central crack starts to open near the crest and on the downstream face of the cohesive core, developing downward. At a given moment, this crack splits into two diagonal cracks as shown in Figure 11a.

The second test, MAIN2 ($w' = 0.52$ m), failed when the second protection slot was removed. In this case, the central vertical crack is almost undetectable. On the contrary, the two diagonal cracks that develop downward and into the base corners of the unprotected area begin near the crest (Figure 11b).

The third test, MAIN3 ($w' = 1.40$ m), also failed when the second protection slot was removed. In this case, the 'diagonal' cracks do not diverge from a central vertical crack. They rather initiate in the crest, far from the vertical axis of the breach and do not necessarily converge to the base corners of the breach. These cracks could also take a 'vertical' trajectory parallel to the wall of the breach. (Figure 11c). The central block is much wider than in the previous two tests.

In the extra test, MAIN3+ ($w' = 1.40$ m), for one level of unprotection, the reservoir level was increased until failure. The mechanisms are the same as in the MAIN3 but, in this case, the cracks that form on the downstream face of the cohesive core were more inclined (Figure 11d). This test was also recorded from the upstream side, allowing us to record the cracks forming on the upstream face. A horizontal crack develops at the base of the breach. This is actually a U-shaped crack as it does not develop horizontally from side to side of the entire width of the unprotected area. Near the lateral walls of the breach, the crack is almost vertical (Figure 11e).

In all tests, the geometry and dimensions of the breach formed in the cohesive core were roughly the same as those of the corresponding unprotected area.

## 4. Discussion

### 4.1. Compaction Tests

Other authors [51] have reported different types of compaction curves: (i) typical single peak curves, (ii) $1^{1/2}$ peak curves, (iii) double peak curves, and (iv) curves without distinct optimal moisture content or oddly shaped curves. According to these authors, the properties of clay, which are determined by the physicochemical characteristics of its constituents, were found to affect the shape of the compaction curves. Montmorillonite, the main constituent of bentonite clay, affects the shape of the compaction curve when its content exceeds 15%. Samples with dominant percentages of sand and the remaining portion of 3-layered silicates (illite or montmorillonite), resulted in $1^{1/2}$ peak curves with a unique peak but in which the trend for low moisture contents seem to lead to a second one. In fact, these curves usually present a very high dry density at zero moisture, a unique characteristic of some of the sandy samples tested by these authors. On the other hand, highly cohesive samples (LL > 100) with more than 50% of montmorillonite usually yield oddly shaped curves without an optimal moisture content. Some correlations between the Attenberg limits are reported in the state of the art [52]. For the bentonite contents managed in this work, 18% and 31%, we obtain values of LL of 100 and 178, respectively.

From a general perspective, the compaction curves obtained in our study seem to be oddly shaped curves without an optimal moisture content. However, we could also derive some resemblance with the $1^{1/2}$ peak curves. For both contents of bentonite, we obtain very high dry densities for very low moisture contents. On the other hand, for a bentonite content of 18%, a zone with a relative maximum can be deduced from the tests P8, P9, and P10 (Figure 7a). This relative maximum dry density is detailed in Figure 12. The same seems to occur between the tests P2 and P3 performed on the samples with a

bentonite content of 31% (Figure 7b), but in this case we should have tested at least two more moisture contents in between.

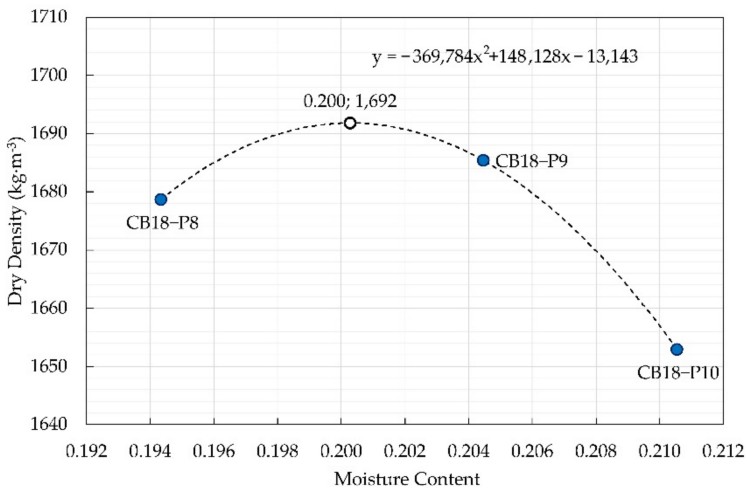

**Figure 12.** Relative maximum dry density and optimal moisture content (white dot) for mixtures with 18% bentonite content.

The compaction tests allowed one to narrow the range of mixtures that could be prepared at the Hydraulics Laboratory for the construction of the cohesive cores. This range, as detailed in previous sections, is roughly that shown in Figure 12, that is, those mixtures varying between the CB18-P8 ($\omega$ = 19.4%) and CB18-P4 ($\omega$ = 22.4%). Mixtures with 31% of bentonite content are not feasible on this scale given their high plasticity.

*4.2. Tools for the Control of Compacticion of the Physical Models*

We emphasize that the results of the geotechnical standardized tests are the ones that should be used to know the strength of compacted soil samples and not those strength values obtained with the Penetrometer or the Pocket Shear Vane Tester. These were only used to control the compaction during the construction of the physical models.

4.2.1. Penetrometer Measurements

Regarding the Penetrometer measurements (Figure 8a), it was observed that the size of the plunger affected the measurements of the soil strength to penetration ($c_p$). By performing one-sided *t*-Student tests between the two groups of measurements (one measurement using the 15 mm plunger on the upper Proctor face and three measurements using the 10 mm plunger also on the upper face assuming the same variance on both groups of measurement) on the compaction tests CB18-P4 and CB18-P10, we obtained *p*-values of 0.063 and 0.108 for CB18-P4 and CB18-P10, respectively. So, for a confidence interval of roughly 90%, we can accept the alternative hypothesis that strength measurements performed with the 15 mm plunge are lower than those performed with the 10 mm plunge.

Therefore, assuming that smaller plunges result in higher soil strength, we could also expect higher strength when measuring with the 6.4 mm plunge. That was precisely what was observed in the CB18-P8 where a disproportional strength was measured (Figure 8a) when it should be, in theory, between the CB18-P9 and CB18-P10 strengths (because the dry density is in between the dry densities of CB18-P9 and CB18-P10).

So, to estimate the penetrometer strength for the relative maximum dry density $\gamma_{d,max}$ = 1692 kN·m$^{-3}$, we have tried two different regression fittings using the least squares method: a logarithmic function fitted to all compaction tests except CB18-P8 and a quadratic function using the tests CB18-P4, CB18-P9 and CB18-P10. In the end, we decided to use the estimate from the quadratic regression curve since it results in a strength value of $c_p$ = 777.3 kPa higher than the rest of the tests (solid red circle in Figure 8a), as expected since we are calculating the strength to penetration for the maximum dry density.

#### 4.2.2. Shear Vane Measurements

Regarding the Humboldt H-4212MH Pocket Shear Vane Tester (Figure 8b), except for the CB18-P5 compaction test where the small vane was used, in the rest of the tests the undrained shear strength ($s_u$) was obtained using the standard size vane (medium size). No comparison was performed between different vane measurements as we did with the Penetrometer, so we cannot assure that there is also a dependency of the soil strength on the size of the vanes. What we observed was that the CB18-P5 strength detaches from the rest of the tests, so we could expect some dependency, although we cannot prove it. Taking this into account, we fitted a quadratic regression function to all tests excluding the results of the CB18-P5 compaction test, to estimate the undrained shear strength for the maximum dry density $\gamma_{d,max}$ = 1692 kN·m$^{-3}$.

The strength measurements performed on CB18-P4, CB18-P8, CB18-P9, and CB18-P10 varied in a wide range, being difficult to discern among these tests. If we compare CB18-P9 with CB18-P8 and CB18-P10 using two-sided *t*-Student tests assuming the same variance for each group of measurements, we obtain *p*-values of 0.235 (CB18-P9 vs. CB18-P8) and 0.960 (CB18-P9 vs. CB18-P10). In both cases, we must retain the null hypothesis, so it is statistically impossible to state that there is a difference between these groups of measurements.

In any case, being expected to have a growing shear vane strength with the dry density of the soils (trend observed in all tests except for CB18-P9), we decided to use the quadratic regression curve to estimate the shear vane strength $s_u$ = 561.1 kPa for the maximum dry density.

#### 4.3. Undrained Unconfined Shear Strength

Regarding Phase I of the Simple Compression set of geotechnical tests, the moisture content of the CB18-P4 soil mixture turned out to be 0.9% higher than expected (Figure 13a). This was the reason why this test was repeated. The moisture of the repeated soil mixture was much closer to its expected value, only varying about 0.2%.

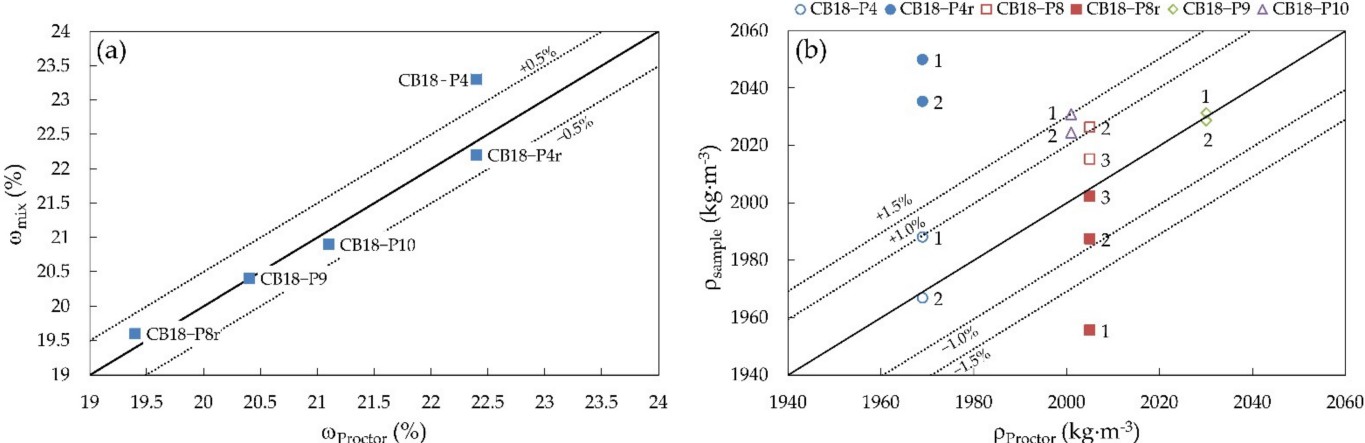

**Figure 13.** Phase I Simple Compression samples compared to the standard Proctor samples. (**a**) Moisture content of the soil mixtures (ωmix) prepared for the compaction of the Simple Compression Phase I samples versus the target moisture content (ωProctor); (**b**) Apparent density of the Simple Compression Phase I samples (ρsample) versus the target density (ρProctor); values refer to the number of the sample.

Despite these differences in the moisture content, the density of the samples originally tested turned out to be much closer to what was expected, that is, much closer to the standard Proctor density, as can be seen in Figure 13b. The density of the original samples varied in a range of ±1.0% from the standard Proctor density (white circles), while the repeated samples were on average 3.7% denser (solid blue circles). These results validate

the differences in the unconfined shear strength, since the original soil samples were on average 8.35 kPa weaker than the repeated samples. This results in a 28% and 39% relative difference in relation to the strengths of the original and repeated compacted soil samples strengths, respectively.

The tests performed on the CB18-P8 samples were repeated because we later found that the scale used to weigh the soil materials for the preparation of the mixture had not previously been tared. Therefore, we do not have the moisture content of the mixture used for the preparation of the original samples. However, we know that they were slightly denser than expected (white squares in Figure 13b).

The CB18-P8 repeated samples were all less dense than expected (solid red squares), although they were prepared with a soil mixture with approximately the expected moisture content (Figure 13a). Except for sample no1, the densities of the CB18-P8 compacted samples varied in a range of ±1% around the expected value. The unconfined shear strength of sample no1 was lower than the rest (Table 6). The unconfined shear strength of these samples (excluding sample no1) was on average 24.90 kPa ± 3.97 kPa (one standard deviation).

The other soil mixtures, CB18-P9 and CB18-P10, also had roughly the expected moisture content (Figure 13a). Although compacted samples prepared with mixture CB18-P9 also had densities around the expected value (white diamonds), samples prepared with mixture CB18-P10 did not (white triangles). They were slightly denser than expected, roughly around 1.0 and 1.5% denser (Figure 13b). If we compare samples no1, both with the same density as can be seen in Table 6, the one prepared with mixture CB18-P9 was stronger than the one prepared with CB18-P10, with an unconfined shear strength around 2.95 kPa higher. On average, samples prepared with the mixture CB18-P9 had an unconfined shear strength of 3.34 kPa higher than those prepared with the mixture CB18-P10. They were also on average denser, as they should be.

If we plot all these results on a single chart we obtain Figure 14, where we have the relationship between the density of the compacted samples with their unconfined shear strength. We can combine this figure with Figure 8 to estimate the undrained shear strength of the compacted cohesive cores constructed at the Hydraulics Laboratory.

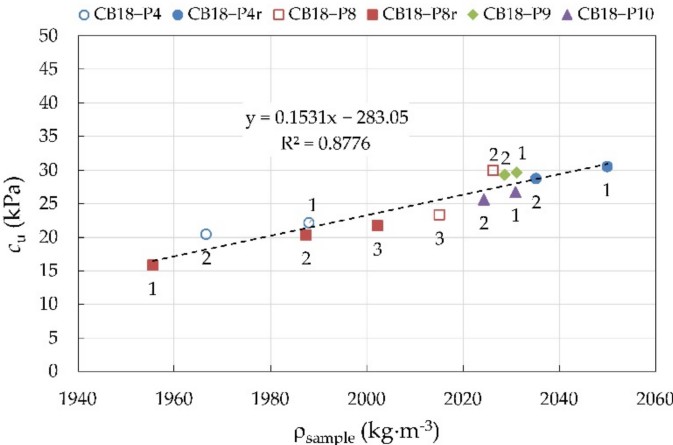

**Figure 14.** Relation between the compacted soil sample's density and the unconfined shear strength. Values refer to the number of the sample.

Phase II of the Simple Compression set of tests was designed to evaluate the evolution of the unconfined shear strength with time. The CB18-P8 mixtures used to prepare the soil samples tested 7 and 28 days after compaction had slightly higher moisture contents than expected, 0.3% and 0.6%, respectively. The resulting soil samples were all denser than expected. The 7-day samples were on average 0.8% denser ± 0.3% (one standard deviation), while the 28-day samples were 1.7% ± 0.2%.

However, the unconfined shear strength was roughly the same, 39.64 kPa ± 1.82 kPa for 7-day samples and 39.95 kPa ± 0.73 kPa for the 28-day samples. We can conclude that

the unconfined shear strength reaches its maximum peak somewhere between 1 and 7 days of maturation and stabilizes after this time. The unconfined shear strength of the samples increased on average by 14.90 kPa, 1.6 times greater than their strength after 1 day of maturation.

### 4.4. Undrained Direct Shear Strength

Except for the CB18-P4 samples no3 and no3r (repeated), which were compacted with a moisture content 3.4% higher than expected, the rest were all compacted with a moisture content within a range of ±0.5% (Figure 15a). Regarding densities, also except for the CB18-P4 samples (circles) whose densities varied in a wider range of roughly ±1.5%, the rest of the samples had densities quite close to their expected value (Figure 15b).

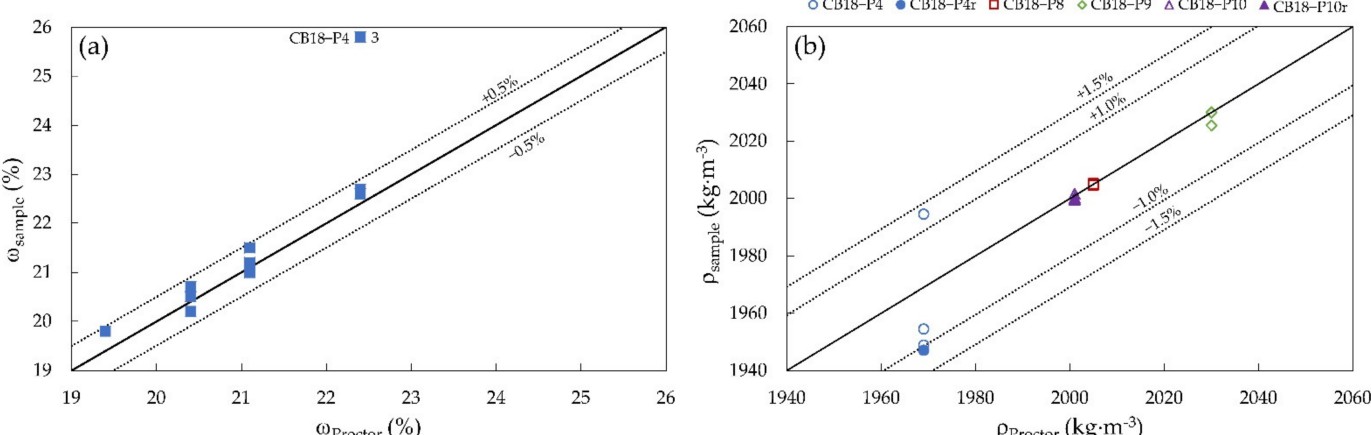

**Figure 15.** Direct Shear samples compared to the standard Proctor samples. (**a**) Moisture content of the Direct Shear samples ($\omega_{sample}$) versus the target moisture content ($\omega_{Proctor}$); (**b**) Apparent density of the Direct Shear samples ($\rho_{sample}$) versus the target density ($\rho_{Proctor}$).

The third sample compacted with the CB18-P4 mixture was repeated because the maximum shear stress was not consistent with what should be expected, that is, if the normal stress increases, the shear stress should also increase. This occurred for all probes except for CB18-P4 samples no3 and no3r (Table 8), so, in this case, these readings were excluded and the strength parameters were estimated only with the first two samples.

The low shear stress of these samples could be related to the excess moisture content and not to differences in the density of the samples, as higher moisture values result in more plastic materials. CB18-P4 samples no2 and no3r have roughly the same density but, despite that, no2 is 1.4 times more resistant than no3r. Additionally, from another point of view, samples no3 and no3r have the same moisture content, but the first is 2.4% denser. In the end, they resulted in roughly the same maximum shear stress.

As mentioned at the beginning of this section, the reason for repeating CB18-P4 sample no3 was related to the excessive moisture content. Nevertheless, this problem was not solved when repeating the test. Something could (not necessarily) have been wrong during these tests that we did not perceive when performing them. This issue could be related, for example, to a wrongly tared scale during the preparation of the mixture.

The CB18-P10 sample no3 was also repeated because although more resistant than the other two samples, its resistance was lower than expected. Something could have been wrong during the performance of the test, so we decided to repeat it. Not the moisture of the samples nor their densities appeared to be the source of this deviation. Unfortunately, by repeating the test we did not observe any difference in the maximum shear stress, which was roughly the same (Table 8). A possible source of error could be related to not having removed particles greater than 1/10 of the sample height, as detailed by the standard. However, the fact that the repeated test resulted approximately in the same shear stress

as the original makes it more improbable that this could be the actual reason for the low shear stresses. Maybe the problem was never with sample no3, but with some of the first two. For example, sample no2 could have resulted in a shear stress higher than expected or sample no1 in a shear stress lower than expected.

*4.5. Displacements of the Cohesive Core*

We would like to emphasize that the removal of protection slots was not instantaneous. Some time passes from the moment we start unscrewing the nuts. In Figure 10f we can see that the displacements start to increase just before the removal of the second protection slot. This is precisely the moment we start to unscrew the wing nuts. This process can be easily observed in the video attached to this manuscript. For future tests, this system should be improved to obtain a faster way of removing the protection slots.

The recorded displacements show an apparent 'plastic' behavior of the cohesive cores, that is, once reached a certain displacement as a consequence of the reservoir level rising, this displacement did not retreat when the reservoir level dropped. This behavior can be observed in Figure 10b where after the removal of the first protection slot, the displacements grew and stabilized even though the reservoir elevation dropped 0.016 m (1.6% of the height of the core). It should be noted that this is probably not a real behavior of the core but simply a limitation of the displacements measuring system as it only allowed recording the core moving downstream. The needle and the mirror were not connected.

This last observation could be related to the delay of the displacements in response to a change in the hydraulic load. This can be clearly observed in Figure 10d. After removing the first slot, the reservoir level drops and when it rises again the displacements need some time to respond. This is probably the result of the needle not being connected to the mirror. When the level of the reservoir drops, the core presumably retreats without being recorded. When it rises again, there is a space in between the needle and the mirror that results in the observed delay of the displacements. The cohesive cores had an elastic behavior for small displacements. That was observed during construction of the cohesive cores. They responded to small pushes but easily recovered their original position. Other possible explanation for this delay could be related to inertial forces.

The MAIN1 and MAIN2 tests had both the same theoretical conditions (breach width of 0.52 m), but MAIN2 deformed roughly three times more than MAIN1 after removing the first protection slot (Figure 10b,d). The reservoir level does not seem to be the cause of this difference as it was higher in MAIN1 than in MAIN2. It could be related to the degree of compaction. If we analyze Figure 9b,c where we present the compaction control of the tests MAIN1 and MAIN2, respectively, we can observe that the cohesive core tested in MAIN1 has higher resistance to penetration than MAIN2, 914 kPa versus 845 kPa, respectively. However, MAIN1 has a lower strength to shear than MAIN2, 508 kPa versus 535 kPa. If the source for the differences in the displacements of both cohesive cores is the degree of compaction, then the penetrometer delivers more reliable measurements than the shear vane. A third source for these differences could be the robustness of the aluminum metallic support, but even this does not fit the observations because MAIN1 had higher displacements than test MAIN2 before taring the displacements to zero (Figure 10a,c).

*4.6. Size and Geometry of the Breach*

As observed during the tests, the size and shape of the breach was roughly the same as the unprotected area. Although almost trivial, this is a very interesting observation. From this we can derive that the shape and dimensions of the breach formed on the rockfill downstream shoulder could control the shape and dimensions of the breach formed on the cohesive central core. This is particularly interesting taking into account that the width of the breach formed on the rockfill downstream slope is highly dependent on the value of the slope [27].

The objective behind testing two different widths was to see how this change could affect the depth of the breach, as it would be expected that wider unprotections would

result in the collapse of the core for a lower depth. Tests MAIN1 and MAIN2 had both the same theoretical conditions (breach width $w' = 0.52$ m), but MAIN1 failed when the third horizontal slot was removed and MAIN2 when removing the second one. These differences are possibly the result of some methodological gaps related, for example, to the compaction of the cohesive cores, as discussed in Section 4.5, or even with the decision of removing a given protection slot that will be discussed in the following Section 4.7. In the test MAIN3 a wider breach was tested and the cohesive core failed when removing the second slot.

The number of tests is clearly insufficient to overcome the identified methodological gaps, so we cannot state that, for the conditions of this set of tests, wider unprotection of the core results in less deep breaches. For future tests, one possible solution could be to increase the precision of the tests by reducing the height of each horizontal protection slot.

### 4.7. Methodological Considerations

The relation between the reservoir level and the displacements of the cohesive cores presented in Section 3.6 reveal some fragilities of the testing procedure, mainly regarding the criteria to define whether to remove or maintain a given protection slot. Before the tests, we did not know about the degree of sensitivity of the cohesive cores to small variations in the reservoir level. The testing protocol was defined assuming that the reservoir would be constant throughout the entire test and that the displacements of the core would be only the result of removing a given protection slot. However, these tests revealed that maintaining a constant water level was very difficult, leading to water loss and reservoir dropping, and that the cohesive cores were very sensitive to small variations of the water level. The combination of these two observations made it very difficult to discern during the performance of the tests whether a displacement was the result of the water level or the degree of unprotection. We will use test MAIN1 to discuss this issue.

As mentioned in the test protocol detailed in Section 2.6, the decision to remove or maintain a protection slot was based on the displacements recorded 10, 100, and 1000 s after the removal. In test MAIN1, the second slot was removed after 1000 s because the displacements had stopped after removing the first slot (Figure 10b). The core stopped moving because the reservoir level dropped. The best decision would have been to increase the reservoir level to its position and wait another 1000 s.

After removing the second slot, because we observed displacements between $t = 100$ s and $t = 1000$ s, we decided to keep this level of unprotection for another 1000 s. However, although we have also observed displacements between $t = 1000$ s and $t = 2000$ s, we have decided to remove the third slot because we assumed that from a visual point of view the overall displacements were slowing down. After analyzing the results, we realized that the displacements were once again the result of the reservoir oscillations. This is not so critical as in the first slot because if the reservoir would have been maintained constant, the core would certainly stop moving downstream, and the decision to remove the second slot would not be so far from reality. If we have decided not to remove it, the test would have taken around one hour and twenty minutes to fail (5000 s), assuming that it would fail when the total displacements reached 0.026 m, 5% of the horizontal unprotection length (0.52 m). For this calculation we used the average velocity of $2.74 \times 10^{-6}$ m·s$^{-1}$ between $t = 100$ s and $t = 2000$ s. If we compare the maximum theoretical displacements with the total displacements observed when the third slot was removed (0.0123 m), we obtain a factor of safety of 2.1.

For future tests, the extreme sensitivity of the cohesive cores should be taken into account in order to redefine the testing procedure as much as possible to exclude this source for bias.

### 4.8. Time to Failure

Another interesting point of discussion is the definition of the time for failure. Any mechanical model designed to evaluate the stability of a cohesive core [30–32] does not take time as a variable. Assuming that the cores deform with constant velocity for any

reason, it would be expected that we would reach failure if we had waited the necessary amount of time. Assuming that these mechanical models predict failure for any given degree of unprotection but the times for occurrence are extremely large, for example, orders of magnitude of months or years, in practice this core would not fail as we would have had time to fix the damages inflicted on the rockfill downstream slope.

## 5. Conclusions

The main conclusions of this experimental campaign were as follows:

- The unconfined shear strength ($c_u$) of the compacted sand-bentonite mixtures was both density and time-dependent. It grew with the increase in both of these variables. Regarding time, these compacted samples reached their maximum strength before 7 days of curation and were around 60% stronger than the samples tested after 1 day of curation. No differences were observed between 7 and 28 days of curation.
- The elastic modulus ($E_u$) obtained from the unconfined Simple Compression Tests was also time-dependent. Its value increased 59% on average from 3.9 MPa $\pm$ 0.8 MPa (one standard deviation) in samples tested after 1 day of curation to 6.3 MPa $\pm$ 0.8 MPa after 7 days of curation. No differences were observed between 7 and 28 days of curation.
- The cohesive central cores were extremely sensitive to small oscillations of the reservoir level. Even risings of just a few millimeters produced displacement of the crest in the downstream direction.
- The cohesive cores behaved as rigid bodies, like concrete slabs with three fixed sides and one free. Cracks formed on the downstream face developed until three independent blocks were formed, two of them rotating around the 'vertical' axes located in the lateral walls of the breach and one around the 'horizontal' axis located at the base of the breach.
- The shape and dimensions of the breach formed on the cohesive cores had roughly the same shape and dimensions as the unprotected area.
- The objective behind testing two different widths was to see how this change could affect the depth of the breach, as it would be expected that wider unprotections would result in the collapse of the core for a lower depth. For the conditions of this set of tests and its limitations, we cannot state that a wider unprotection of the cohesive core results in less deep breaches.

**Author Contributions:** Conceptualization, R.M.-A., M.Á.T. and R.M.; methodology, R.M.-A. and J.P.; software, R.M.-A.; validation, M.Á.T. and R.M.; formal analysis, R.M.-A.; investigation, R.M.-A.; resources, M.Á.T. and R.M.; data curation, R.M.-A. and J.P.; writing—original draft preparation, R.M.-A.; writing—review and editing, M.Á.T., R.M. and J.P.; visualization, R.M.-A. and J.P.; supervision, M.Á.T. and R.M.; project administration, R.M.-A., M.Á.T. and R.M.; funding acquisition, M.Á.T. and R.M. All authors have read and agreed to the published version of the manuscript.

**Funding:** This research was funded by the Spanish Ministry of Economy and Competitiveness, grant number RTC-2016-4967-5 (Project HIRMA—*Desarrollo y validación de una aplicación para la determinación del hidrograma de rotura de presas de materiales sueltos, a partir de la configuración geomecánica particular*).

**Data Availability Statement:** The results presented in this paper are part of the Ph.D. Thesis of the first author (R.M.-A.) that can be downloaded with the following link: https://oa.upm.es/67371/ (accessed on 5 October 2022).

**Acknowledgments:** We would like to thank José Luis Orts, Raffaella Pellegrino, and Gloria Cachaza for their help during the laboratory tests performed at the Hydraulics Laboratory of the *E.T.S.I de Caminos, Canales y Puertos* of the *Universidad Politécnica de Madrid*. We would also like to thank Cristina Fonollá from the Geotechnics Laboratory of the *E.T.S.I de Caminos, Canales y Puertos* of the *Universidad Politécnica de Madrid* for the performance of the majority of the geotechnical characterization tests. Finally, we would like to thank Claudio Olalla, Rafael Jiménez, and Ignacio Tejada for their consultancy on geotechnical issues crucial to defining the methodology used for testing the cohesive central cores failure.

**Conflicts of Interest:** The authors declare no conflict of interest. The funders had no role in the design of the study; in the collection, analyses, or interpretation of data; in the writing of the manuscript, or in the decision to publish the results.

## Nomenclature

The following symbols and acronyms are used in this paper:

| | |
|---|---|
| $c_{\mathrm{p}}$ | Soil strength to penetration measured with the Geotester Pocket Penetrometer kit (fundamental units M·L$^{-1}$·T$^{-2}$) |
| $c_{\mathrm{u}}$ | Unconfined shear strength calculated from the Simple Compression Test (fundamental units M·L$^{-1}$·T$^{-2}$) |
| $c_{\mathrm{uu}}$ | Cohesion of the compacted soil samples obtained from the unconsolidated undrained direct shear strength (fundamental units M·L$^{-1}$·T$^{-2}$) |
| $C_{\mathrm{u}}$ | Coefficient of uniformity, the ratio D60/D10 (dimensionless) |
| D10 | Sieve size passing 10% of the particles (fundamental units L) |
| D50 | Sieve size passing 50% of the particles (fundamental units L) |
| D60 | Sieve size passing 60% of the particles (fundamental units L) |
| $E_{\mathrm{u}}$ | Elastic modulus calculated from the Unconfined Simple Compression Tests (fundamental units M·L$^{-1}$·T$^{-2}$) |
| $H$ | Cohesive core height (fundamental units L) |
| $H_{\mathrm{r}}$ | Reservoir water elevation from the flume base (fundamental units L) |
| $I_{\mathrm{b}}$ | Width of the cohesive core base (fundamental units L) |
| $I_{\mathrm{c}}$ | Width of the cohesive core crest (fundamental units L) |
| MAIN | Main Laboratory Experiments |
| $p_{\mathrm{S:B}}$ | Sand-bentonite proportion in weight (dimensionless) |
| PRELIM | Preliminary Laboratory Experiments |
| $q_{\mathrm{u}}$ | Unconfined compressive strength obtained from the Simple Compression Test (fundamental units M·L$^{-1}$·T$^{-2}$) |
| $s$ | Real displacements along the flume longitudinal axis (fundamental units L) |
| $s_{\mathrm{u}}$ | Undrained shear strength measured with the Humboldt H-4212MH Pocket Shear Vane Tester (fundamental units M·L$^{-1}$·T$^{-2}$) |
| $S$ | Amplified displacements along the flume walls (fundamental units L) |
| $t_{\mathrm{failure}}$ | Time for the failure of the compacted soil samples (fundamental units T) |
| $w'$ | Width of the cohesive core unprotected area (fundamental length L) |
| $W$ | Generic apparent/moistured mass of a given amount of soil (fundamental units M) |
| $W_{\mathrm{B}}$ | Apparent/moistured mass of bentonite (fundamental units M) |
| $W_{\mathrm{B,d}}$ | Mass of dry bentonite (fundamental units M) |
| $W_{\mathrm{d}}$ | Generic mass of a dry amount of soil (fundamental units M) |
| $W_{\mathrm{H2O,add}}$ | Mass of water to add to a given sand-bentonite mixture to reach the mixture desired moisture content ($\omega_{\mathrm{goal}}$) (dimensionless) |
| $W_{\mathrm{S}}$ | Apparent/moistured mass of sand (fundamental units M) |
| $W_{\mathrm{S,d}}$ | Mass of dry sand (fundamental units M) |
| $x_0$ | Distance along the flume longitudinal axis between the initial position of the needle and the laser pointer (fundamental units L) |
| $X_0$ | Distance along the flume's right wall between the laser pointer and the initial position of the projected laser beam point (fundamental units L) |
| $\alpha$ | Angle of the original laser beam direction (perpendicular to the flume) with the new position of the rotating mirror's arm (degrees) |
| $\alpha_0$ | Angle of the original laser beam direction (perpendicular to the flume) with the initial position of the mirror's arm on which the needle leans on (degrees) |
| $\beta$ | Angle of the original laser beam direction (perpendicular to the flume) with the projected laser beam direction after rotation (degrees) |
| $\beta_0$ | Angle of the original laser beam direction (perpendicular to the flume) with the projected laser beam direction before any rotation (degrees) |

| | |
|---|---|
| $\phi_{uu}$ | Friction angle of the compacted soil samples obtained from the unconsolidated undrained direct shear strength (degrees) |
| $\gamma$ | Mirror's rotation angle (dimensionless) |
| $\varepsilon_{failure}$ | Strain/deformation for the failure of the compacted soil samples (dimensionless) |
| $\rho$ | Aparent/moistened density of the compacted soil samples (fundamental units $M \cdot L^{-3}$) |
| $\rho_d$ | Dry density of the compacted soil samples (fundamental units $M \cdot L^{-3}$) |
| $\rho_{Proctor}$ | Apparent/moistened density of the soil mixtures compacted within the Standard Proctor compaction tests campaign (fundamental units $M \cdot L^{-3}$) |
| $\rho_{sample}$ | Apparent/moistened density of a given sample of compacted soil (fundamental units $M \cdot L^{-3}$) |
| $\tau_{max}$ | Maximum shear stress ined from the unconsolidated undrained Direct Shear strength (fundamental units $M \cdot L^{-1} \cdot T^{-2}$) |
| $\omega$ | Moisture content (dimensionless) |
| $\omega_{add}$ | Laking moisture content, i.e., the difference between the desired moisture content ($\omega_{goal}$) and the sand-bentonite mixture initial moisture content ($\omega_{SB,i}$) (dimensionless) |
| $\omega_B$ | Bentonite's moisture content (dimensionless) |
| $\omega_{goal}$ | Desired moisture content for a given sand-bentonite mixture (dimensionless) |
| $\omega_{mix}$ | Moisture content of the soil mixtures before compaction (dimensionless) |
| $\omega_{Proctor}$ | Moisture content of the soil mixtures compacted within the Standard Proctor compaction tests campaign (dimensionless) |
| $\omega_S$ | Sand's moisture content (dimensionless) |
| $\omega_{sample}$ | Moisture content of a given sample of compacted soil (dimensionless) |
| $\omega_{SB,i}$ | Initial moisture content of a sand-bentonite mixture resulting from mixing the sand and the bentonite with their natural moisture contents in a given proportion $p_{S:B}$ (dimensionless) |

# Appendix A

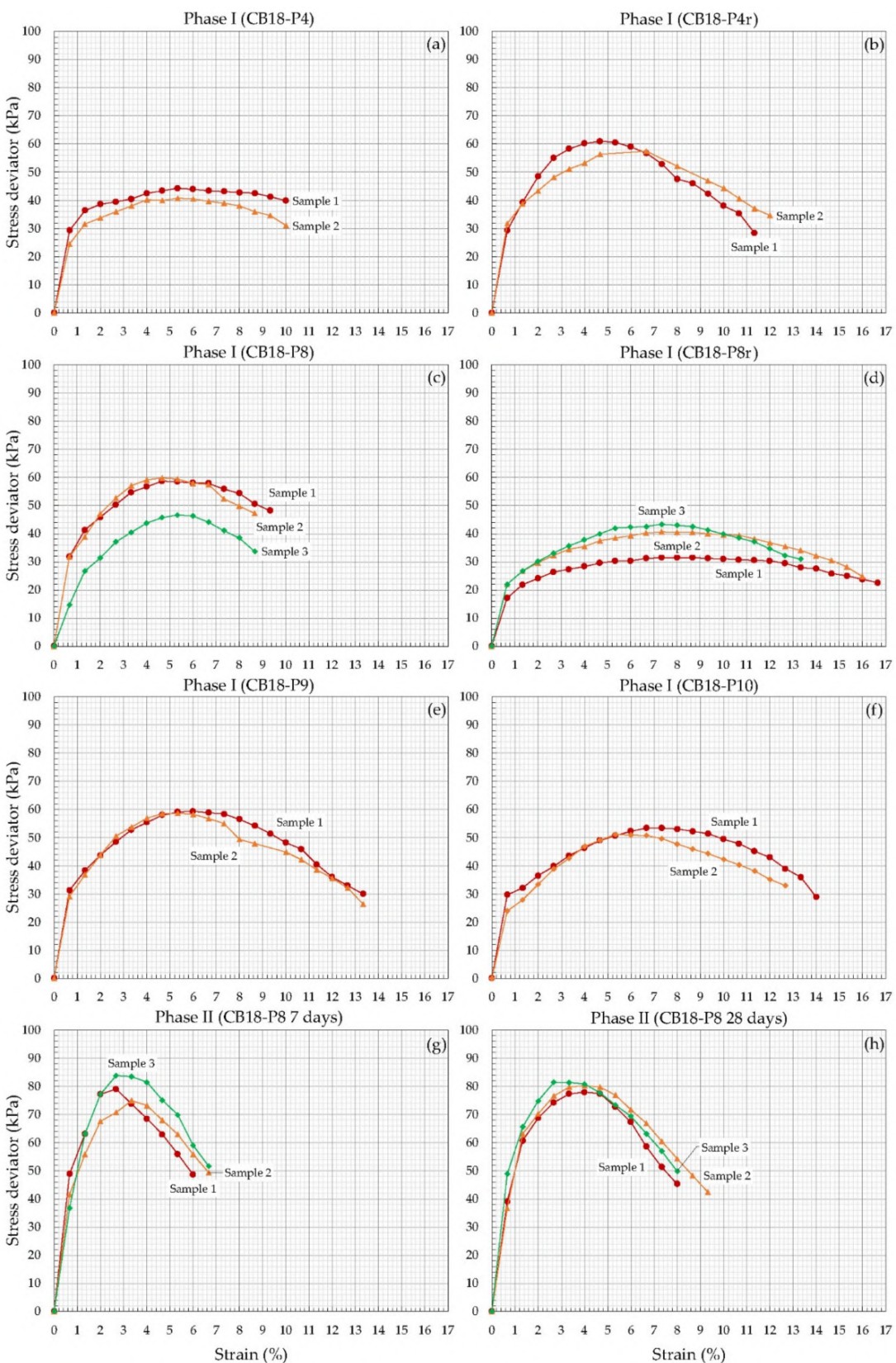

**Figure A1.** Results of the Simple Compression tests. (**a**–**f**) Phase I; (**g**–**h**) Phase II.

## Appendix B

**Table A1.** Displacements of the cohesive central core tested in MAIN1.

| Time (s) | Test Time (hh:mm:ss) | Displacement *s* (mm) | Time (s) | Test Time (hh:mm:ss) | Displacement *s* (mm) |
|---|---|---|---|---|---|
| 3571 | 13:05:05 | 0.0 | 4713 | 13:24:07 | 5.7 |
| 3592 | 13:05:26 | 0.1 | 4717 | 13:24:11 | 5.7 |
| 3594 | 13:05:28 | 0.2 | 4718 | 13:24:12 | 5.9 |
| 3595 | 13:05:29 | 0.3 | 4726 | 13:24:20 | 6.1 |
| 3596 | 13:05:30 | 0.4 | 4736 | 13:24:30 | 6.4 |
| 3597 | 13:05:31 | 0.8 | 4756 | 13:24:50 | 6.6 |
| 3598 | 13:05:32 | 1.1 | 4794 | 13:25:28 | 7.1 |
| 3599 | 13:05:33 | 1.9 | 4848 | 13:26:22 | 7.3 |
| 3613 | 13:05:47 | 1.9 | 4914 | 13:27:28 | 7.6 |
| 3704 | 13:07:18 | 1.9 | 4984 | 13:28:38 | 7.8 |
| 4598 | 13:22:12 | 1.9 | 5070 | 13:30:04 | 8.1 |
| 4670 | 13:23:24 | 1.9 | 5185 | 13:31:59 | 8.5 |
| 4674 | 13:23:28 | 2.0 | 5695 | 13:40:29 | 8.6 |
| 4675 | 13:23:29 | 2.1 | 5799 | 13:42:13 | 9.9 |
| 4677 | 13:23:31 | 2.2 | 5911 | 13:44:05 | 10.2 |
| 4678 | 13:23:32 | 2.3 | 5944 | 13:44:38 | 10.5 |
| 4680 | 13:23:34 | 2.7 | 5974 | 13:45:08 | 10.8 |
| 4681 | 13:23:35 | 2.8 | 5986 | 13:45:20 | 10.9 |
| 4682 | 13:23:36 | 2.9 | 6018 | 13:45:52 | 11.1 |
| 4683 | 13:23:37 | 3.0 | 6354 | 13:51:28 | 11.2 |
| 4684 | 13:23:38 | 3.1 | 6386 | 13:52:00 | 11.4 |
| 4686 | 13:23:40 | 3.2 | 6440 | 13:52:54 | 11.7 |
| 4688 | 13:23:42 | 3.3 | 6495 | 13:53:49 | 12.0 |
| 4689 | 13:23:43 | 3.4 | 6531 | 13:54:25 | 12.1 |
| 4690 | 13:23:44 | 3.5 | 6555 | 13:54:49 | 12.3 |
| 4694 | 13:23:48 | 3.8 | 6703 | 13:57:17 | 12.3 |
| 4697 | 13:23:51 | 4.3 | 6846 | 13:59:40 | 13.2 |
| 4698 | 13:23:52 | 4.6 | 6848 | 13:59:42 | 13.4 |
| 4699 | 13:23:53 | 4.9 | 6849 | 13:59:43 | 13.6 |
| 4701 | 13:23:55 | 5.0 | 6850 | 13:59:44 | 14.5 |
| 4703 | 13:23:57 | 5.1 | 6852 | 13:59:46 | 15.1 |
| 4705 | 13:23:59 | 5.2 | 6853 | 13:59:47 | 16.6 |
| 4707 | 13:24:01 | 5.4 | 6854 | 13:59:48 | 17.8 |
| 4710 | 13:24:04 | 5.6 | 6855 | 13:59:49 | 19.3 |

**Table A2.** Displacements of the cohesive central core tested in MAIN2.

| Time (s) | Test Time (hh:mm:ss) | Displacement *s* (mm) | Time (s) | Test Time (hh:mm:ss) | Displacement *s* (mm) |
|---|---|---|---|---|---|
| 2508 | 13:10:00 | 0.0 | 3763 | 13:30:55 | 8.3 |
| 2585 | 13:11:17 | 1.3 | 3882 | 13:32:54 | 8.6 |
| 2611 | 13:11:43 | 4.1 | 3934 | 13:33:46 | 9.1 |
| 2614 | 13:11:46 | 4.2 | 3959 | 13:34:11 | 9.2 |
| 2616 | 13:11:48 | 4.3 | 3961 | 13:34:13 | 9.3 |
| 2621 | 13:11:53 | 5.3 | 4559 | 13:44:11 | 9.3 |
| 2622 | 13:11:54 | 5.4 | 4630 | 13:45:22 | 9.4 |
| 2624 | 13:11:56 | 5.8 | 4633 | 13:45:25 | 10.0 |
| 2625 | 13:11:57 | 6.5 | 4634 | 13:45:26 | 10.6 |
| 2627 | 13:11:59 | NA | 4636 | 13:45:28 | 11.3 |
| 2637 | 13:12:09 | 6.5 | 4637 | 13:45:29 | 11.4 |
| 2727 | 13:13:39 | 6.5 | 4638 | 13:45:30 | 11.8 |
| 2921 | 13:16:53 | 6.5 | 4639 | 13:45:31 | 12.0 |
| 2924 | 13:16:56 | 6.8 | 4641 | 13:45:33 | 12.1 |
| 2970 | 13:17:42 | 7.0 | 4643 | 13:45:35 | 13.9 |

**Table A2.** *Cont.*

| Time (s) | Test Time (hh:mm:ss) | Displacement *s* (mm) | Time (s) | Test Time (hh:mm:ss) | Displacement *s* (mm) |
|---|---|---|---|---|---|
| 3012 | 13:18:24 | 7.3 | 4644 | 13:45:36 | 16.1 |
| 3053 | 13:19:05 | 7.5 | 4646 | 13:45:38 | 17.9 |
| 3071 | 13:19:23 | 7.6 | 4647 | 13:45:39 | 19.4 |
| 3093 | 13:19:45 | 7.8 | 4649 | 13:45:41 | 20.5 |
| 3560 | 13:27:32 | 7.8 | 4650 | 13:45:42 | 21.0 |
| 3627 | 13:28:39 | 7.8 | | | |

**Table A3.** Displacements of the cohesive central core tested in MAIN3.

| Time (s) | Test Time (hh:mm:ss) | Displacement *s* (mm) | Time (s) | Test Time (hh:mm:ss) | Displacement *s* (mm) |
|---|---|---|---|---|---|
| 9 | 12:40:00 | 0.5 | 3409 | 13:36:40 | 5.3 |
| 309 | 12:45:00 | 0.5 | 3699 | 13:41:30 | 5.5 |
| 3284 | 13:34:35 | 0.5 | 3754 | 13:42:25 | 5.8 |
| 3287 | 13:34:38 | 0.5 | 3959 | 13:45:50 | 6.0 |
| 3289 | 13:34:40 | 1.0 | 4270 | 13:51:01 | 6.3 |
| 3290 | 13:34:41 | 2.0 | 4309 | 13:51:40 | 6.3 |
| 3292 | 13:34:43 | 3.0 | 4465 | 13:54:16 | 17.3 |
| 3296 | 13:34:47 | 3.5 | 4466 | 13:54:17 | 19.5 |
| 3299 | 13:34:50 | 4.0 | 4467 | 13:54:18 | 20.3 |
| 3302 | 13:34:53 | 4.3 | 4468 | 13:54:19 | 21.5 |
| 3309 | 13:35:00 | 4.5 | 4469 | 13:54:20 | 23.5 |
| 3311 | 13:35:02 | 4.8 | 4470 | 13:54:21 | 24.8 |
| 3312 | 13:35:03 | 5.0 | 4471 | 13:54:22 | 26.0 |
| 3319 | 13:35:10 | 5.0 | 4472 | 13:54:23 | 27.5 |

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
