# Peer review of "Structural Failure of the Cohesive Core of Rockfill Dams: An Experimental Research Using Sand-Bentonite Mixtures"

_water, doi:10.3390/w14233966_

Round 1

Reviewer 1 Report

This article is very well done, well structured and answers the journal's theme. 

However I advise the author to rectify the comments that I presented throughout this article, then to update them.

Reviewer 2 Report

The article deals with laboratory tests of vertical wall made of cohesive soil, which is a model of cohesive core in rockfill dam. The soil material was a mixture of synthetic sand and bentonite. The tests were carried out in a flume 1.5m wide, the height of the tested model was ca 0.6 – 1.0m, the impervious element (diaphragm) was supported on the downstream side  by vertical sheet wall, composed of horizontal elements 0.2m high. The research consisted in determining the deformation of the upper edge of the diaphragm depending on the mineralogical composition of the material, the shape of the diaphragm, the height of the support of it, the height of the water in the reservoir or the flow rate above the diaphragm. The authors presented in detail the geotechnical characteristics of soil mixture, the results of the degree of compaction, shear and compressive strength. The construction of the model, the measurement apparatus used and the method of performing the measurements as well as the results obtained have been described in detail.

Critical remarks:

1.     The purpose of the article is to determine the time of deformation leading to the destruction of cohesive core in rockfill dam. At the same time, the research is concerned only with the behavior of a cohesive material diaphragm, which is supported by an aluminum wall, the body of rockfill dam does not appear in the research. The authors, however, do not explain how to relate the results of their study to actual dams and their risk of failures. The results are presented in the form of dimensional charts, so converting them to the dimensions of actual structure is impossible.

2.     The essential research concerns a very thin dam core, the width at the base of which is only ca 10% of the height (MAIN 1 – MAIN 3). The reviewer is not aware of dams whose core from cohesive soil would have such proportions. Such thin core may crack during construction or subsequent settlement of the dam. The authors should indicate which existing dams they modeled on when proposing such core shape.

3.     Proposed material in the form „synthetic sand-bentonite mixtures” is used to prevent seepage in landfills, but is it economically viable to make such core of the dam compared to natural material such as clay?

4.     The research did not assess the core erosion process, please explain why.

5.     The content of bentonite in mixtures for the dam core should be within 0.5-5%. To little will cause the pore to fail to close and the core will be permeable, too much will cause a decrease in shear strength [1]. Please explain why a mix of 18% and 31% bentonite was tested.

6.     The width of the breach significantly affects the deformation of the cohesive core. The study used two widths (unprotected areas), 0.5m and 1.5m. Please clarify whether these widths correspond to the values of breach of historical rockfill dams and have been scaled relative to, for example, height?

7.     The study does not provide values for the seepage coefficient of the mixture.

Minor remarks

1.     Line 57 and the next few – Error!... correct references

2.     In table 8, the value of soil friction angle for CB18-P4 is only 16.1° and is three times smaller than CB18-P8, which is of the same material. Please clarify this.

3.     Line 392 – the tests last maximum of 14h, why is there the value of 1 year as time to failure?

1.     Proiaa' R., Crocea P., Modonia G., Experimental investigation of compacted sand-bentonite mixtures. Procedia Engineering 158 (2016) 51 — 56.

Round 2

Reviewer 2 Report

After the Authors' extensive explanations and changes presented in the paper, I recommend the paper for consideration for publication